# TMEM106B drives lung cancer metastasis by inducing TFEB-dependent lysosome synthesis and secretion of cathepsins

Samrat T. Kundu[1], Caitlin L. Grzeskowiak[2], Jared J. Fradette[1], Laura A. Gibson[1], Leticia B. Rodriguez[1], Chad J. Creighton[3,4], Kenneth L. Scott[2] & Don L. Gibbons[1,5]

Metastatic lung cancer is the leading cause of cancer-associated mortality worldwide, therefore necessitating novel approaches to identify specific genetic drivers for lung cancer progression and metastasis. We recently performed an in vivo gain-of-function genetic screen to identify driver genes of lung cancer metastasis. In the study reported here, we identify TMEM106B as a primary robust driver of lung cancer metastasis. Ectopic expression of TMEM106B could significantly promote the synthesis of enlarged vesicular lysosomes that are laden with elevated levels of active cathepsins. In a TFEB-dependent manner, TMEM106B could modulate the expression of lysosomal genes of the coordinated lysosomal expression and regulation (CLEAR) pathway in lung cancer cells and patient samples. We also demonstrate that TMEM106B-induced lysosomes undergo calcium-dependent exocytosis, thereby releasing active lysosomal cathepsins necessary for TMEM106B-mediated cancer cell invasion and metastasis in vivo, which could be therapeutically prevented by pharmacological inhibition of cathepsins. Further, in TCGA LUAD data sets, 19% of patients show elevated expression of TMEM106B, which predicts for poor disease-free and overall-survival.

[1] Department of Thoracic/Head and Neck Medical Oncology, The University of Texas MD Anderson Cancer Center, Houston, TX 77030, USA. [2] Department of Molecular and Human Genetics, Baylor College of Medicine, Houston, TX 77030, USA. [3] Department of Medicine and Dan L. Duncan Cancer Center, Baylor College of Medicine, Houston, TX 77030, USA. [4] Department of Bioinformatics and Computational Biology, The University of Texas MD Anderson Cancer Center, Houston, TX 77030, USA. [5] Department of Molecular and Cellular Oncology, The University of Texas MD Anderson Cancer Center, Houston, TX 77030, USA. These authors contributed equally: Samrat T. Kundu, Caitlin L. Grzeskowiak. Deceased: Kenneth L. Scott. Correspondence and requests for materials should be addressed to S.T.K. (email: skundu@mdanderson.org) or to D.L.G. (email: dlgibbon@mdanderson.org)

Lung cancer is a deadly disease resulting in the highest lethality rates among all cancers. Like other cancers, metastasis is the primary cause of cancer-related mortality among lung cancer patients[1]. The development of new therapeutic strategies requires a better understanding of the underlying "driver" genetic aberrations responsible for lung cancer metastasis, as such events represent new targets for drug intervention. To identify such driver events, we previously used function-based gain-of-function in vitro and in vivo genetic screens to successfully identify new oncogenes and their activating mutations in melanoma[2], breast cancer[3], and pancreatic cancer[4] among others. Using a similar approach, we recently performed a robust, in vivo positive selection screen to identify individual drivers of lung cancer progression and metastasis[5]. For this, we performed a gain-of-function screen where we identified an enriched list of 217 putative driver genes in lung cancer by cross-species comparison of the genes overexpressed in autochthonous genetically engineered metastatic murine lung tumors and syngeneic lung cancer models, with human copy number amplifications documented by The Cancer Genome Atlas (TCGA). We used a mouse syngeneic lung cancer model, which was developed in our lab to elucidate the molecular mechanisms underlying epithelial-to-mesenchymal transition and metastasis. The model comprises of different cell lines isolated from lung tumors and metastases from genetically engineered $Kras^{LA1/+};p53^{R172H\Delta G/+}$ mice, which vary in their metastatic potential when implanted in syngeneic mice[6–8]. Next, we generated a lentiviral unique-barcoded expression library, utilizing sequence-verified open reading frames (ORFs) of the candidate genes. Using a nonmetastatic syngeneic mouse lung cancer cell line (393P), we generated individually transduced stable overexpressing lines for each candidate gene. We transplanted pools of ORF expressing lines into syngeneic immune-competent 129sv mice, and observed primary tumor growth and occurrence of metastases in lungs and other organs. DNA extracted for primary tumors and metastases was used for barcode sequencing by next-generation sequencing. Metastasis drivers were identified upon enrichment of their specific ORF-associated barcodes in metastases, when compared to total barcode reads in input cells. Among others, we identified TMEM106B as one of the most robust drivers of lung cancer metastasis in vivo.

TMEM106B is a single pass, type-II transmembrane protein reported to localize on cellular lysosomes. While the protein has not been thoroughly studied in the context of cancer, TMEM106B loss-of-function is known to be associated with frontotemporal lobar degeneration, and is also repressed in the brains of Alzheimer's patients[9–12]. More recently it was reported that TMEM106B has a prominent role in regulating lysosome synthesis, size, trafficking, and localization[13,14]. It was also reported that overexpression of TMEM106B in neuronal cells resulted in the activation of the lysosomal stress signaling by translocating the transcription factor TFEB, a member of the MiTF family, into the nucleus and thus up-regulating the coordinated lysosomal expression and regulation (CLEAR) network genes[14]. The MiTF family of transcription factors has also been shown to directly regulate the expression of lysosomal hydrolases, primarily cathepsin K[15,16].

Lysosomes are phospholipid bilayer membrane bound vesicles, ubiquitously present in all cell types and essential for the catabolic clearance and recycling of degraded proteins, macromolecules, dysfunctional organelles, and digestion of extracellular and foreign materials delivered to them by endocytosis, autophagy, and phagocytosis[17–20]. More recently, lysosome functions have been reported to be dramatically changed during cancer progression, where lysosomal volume, composition, and subcellular localization have been altered[21]. Lysosomes are reported to be routed toward the periphery of the cell under multiple stimuli, which are responsible for the onset of different pathological states. For example, during malignant transformation the peripheral juxtaposition of lysosomes with the plasma membrane and its contribution to cell invasion and migration makes this phenomenon of crucial importance for our research[22–24]. There is evidence that shows the elevated expression or activity of TFEB as a positive induction for lysosomal exocytosis. It has also been reported that increased production of lysosomal cathepsin B and L could act as signals for increased lysosome trafficking to the plasma membrane[22,25,26]. Upon localization at the pericellular space, the lysosome are fused to each other before fusing with the plasma membrane, usually as a result of intracellular calcium influx. This final fusion of the lysosomes to the plasma membrane results in the transfer of the luminal sialylated lysosomal membrane protein LAMP1 to the plasma membrane, and also the release of the complete lysosomal content into the extracellular matrix or media.

Cathepsins and other lysosomal hydrolases have been reported to be significantly overexpressed in a number of cancers leading to increased invasion of cancer cells[23,27–29]. In our syngeneic mouse model, we have observed by proteomic profiling that the metastatic cells demonstrated a significantly high level of secreted cathepsins[30], which might drive the hyperinvasive and metastatic phenotypes. Cathepsins secreted into the extracellular matrix (ECM) via lysosomal exocytosis or cell lysis, are the primary effectors of ECM degradation, dissolution of cell–cell adhesion molecules, and activation or initiation of cytokine and chemokine responses that are all critically contributory toward cancer cell invasion and metastasis[24,31–35]. Here, we have identified TMEM106B as a novel and specific driver of lung cancer metastasis. TMEM106B deregulates lysosome function by affecting lysosomal synthesis and exocytosis in lung cancer cells, thus resulting in increased release of lysosomal cathepsins into the extracellular matrix, leading to cellular invasion and metastasis. Therapeutic intervention of TMEM106B activity by inhibiting cathepsins in vivo in a metastatic syngeneic mice model with aloxistatin treatment successfully blocks TMEM106B-mediated metastasis.

## Results

**TMEM106B is a novel driver of invasion and metastasis**. To identify specific genes with a causative role in metastasis, we recently performed an in vivo positive selection screen in which 251 DNA-barcoded cDNAs selected from murine and human genomics data sets were cloned and assessed for metastasis-promoting activity. These cDNAs were delivered to tumorigenic, nonmetastatic 393P murine lung cancer cells subsequently pooled for subcutaneous (SQ) injection into syngeneic mice. Out of the 217 genes screened, we identified 28 potent drivers of lung cancer metastasis (~12% of the screened genes) based on positive enrichment of their associated barcodes in lung metastases. Among the highest metastasis-enriched cDNAs was that encoding TMEM106B, whose associated barcode was present in 10 independent lung metastasis tissue cores from 5 different animals in the screen. To further validate TMEM106B as a driver of in vivo metastasis, 393P cells expressing the top five identified drivers and Mcherry (control) were implanted independently into the flanks of 129SV syngeneic mice, and observed for primary tumor growth and distant metastasis. Though primary tumor sizes for all the candidate lines were not significantly different from each other or control (Fig. 1a), TMEM106B overexpressing cells showed significantly higher numbers of in vivo metastasis compared to Mcherry control cells. Additionally, the mice injected with TMEM106B overexpressing cells demonstrated the highest median, 14.5 (Mcherry = 0, GNAS = 5, TMEM106B = 14.5, ORMDL3 = 4, THRA = 3.5, and

CWF19L2 = 7) and mean, 24.17 (Mcherry = 0.92, GNAS = 8.22, TMEM106B = 24.17, ORMDL3 = 4.4, THRA = 3.5, and CWF19L2 = 8.6) number of metastatic events, which were substantially higher than the other metastasis driver genes identified from our primary in vivo screen (Fig. 1b). The metastatic propensity of TMEM106B-expressing cells can be appreciated from the representative images of the gross lungs and H&E-stained cross

sections, which show a robust increase in the metastatic lung nodules in the mice injected with the TMEM106B cells compared to the Mcherry controls (Fig. 1c). The metastatic phenotype of TMEM106B was a result of the high expression of the TMEM106B transgene in the 393P cells (Fig. 1d), which was corroborated with a significant increase of both migration and invasion in transwell

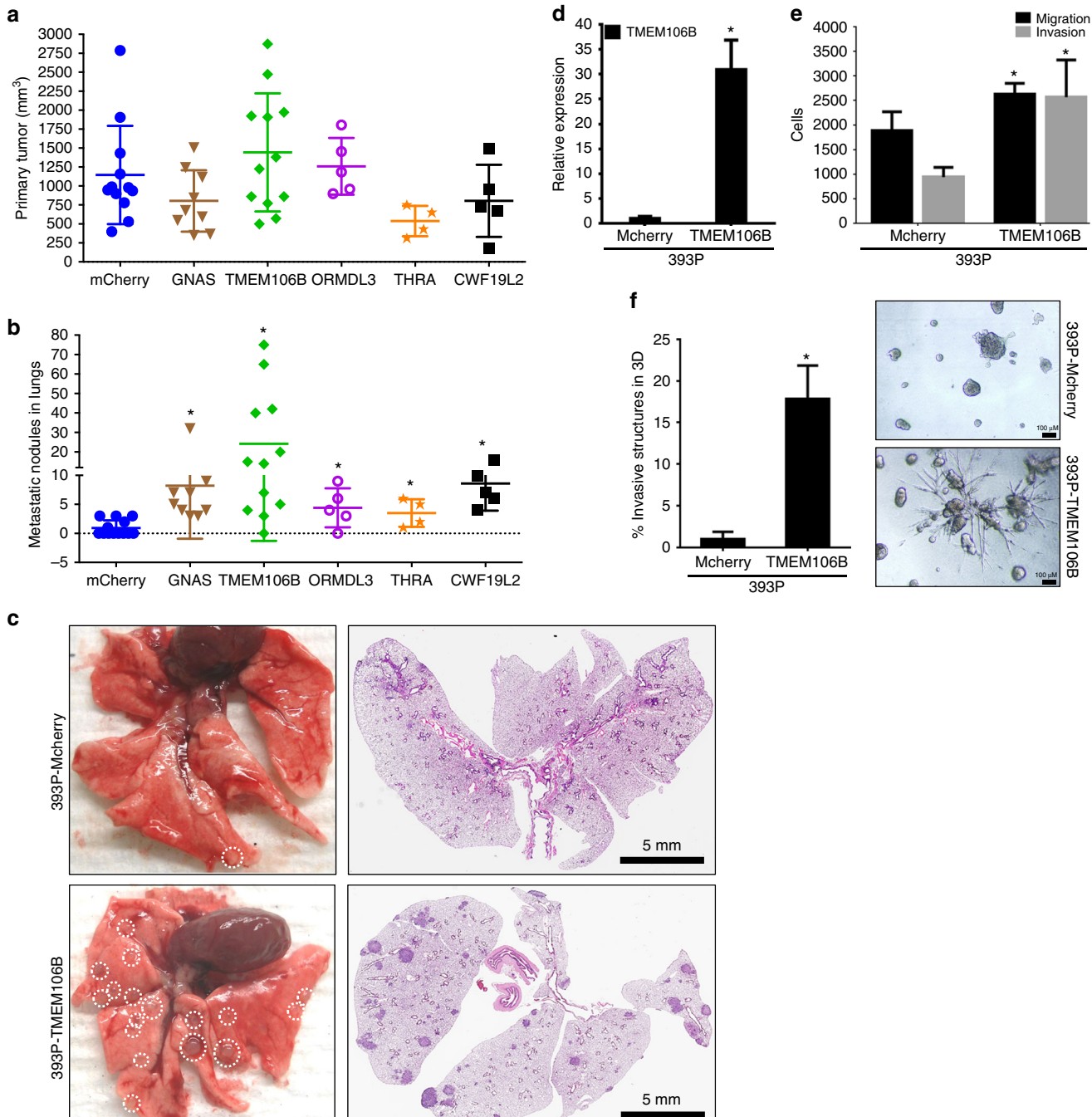

**Fig. 1** TMEM106B is a novel driver of invasion and metastasis in lung cancers. **a** Primary tumor growth for 393P cells overexpressing individual hits (as indicated), identified in the in vivo screen[5] were injected subcutaneously into syngeneic hosts and observed for primary tumor growth and **b** lung metastasis. **c** Representative lungs and their respective H&E sections are shown. Lungs from mice injected with TMEM106B-expressing cells show significantly more incidence of lung metastasis. **d** qPCR analysis for TMEM106B expression in 393P cells overexpressing TMEM106B. **e** Cells overexpressing TMEM106B showing significant increase in migration and invasion compared to Mcherry are indicated. **f** TMEM106B overexpressing cells form significantly more number of invasive spheroids compared to Mcherry when seeded in 3D matrix comprising of matrigel and collagen. All asterisks indicate statistical significance by T test ($n \geq 3$, $^*p \leq 0.05$)

assays (Fig. 1e) and enhanced formation of invasive structures when grown in a 3D matrigel/collagen matrix (Fig. 1f).

**TMEM106B knockdown suppresses invasion and metastasis.** To further validate the functional regulation of metastasis by

TMEM106B, its expression was stably inhibited by shRNAs targeting different segments of the TMEM106B mRNA in the highly metastatic $Kras^{LA1/+};TP53^{R172H\Delta G}$ (KP)[7,36] mouse lung cancer cells, 344SQ and 344LN (TMEM106B-sh1–sh3; Fig. 2a, b and Supplementary Figure 1A), and human A549 cells

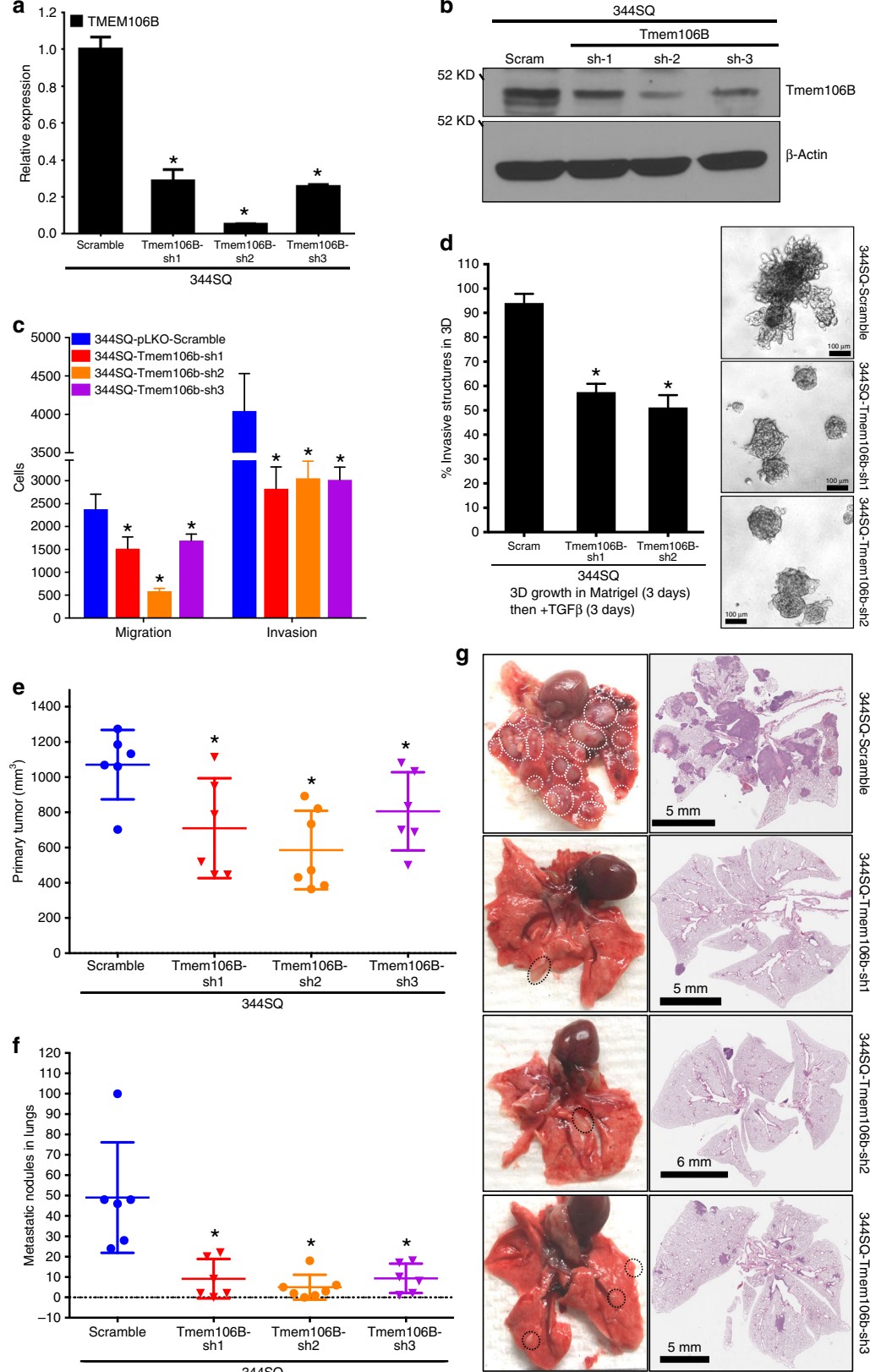

(Supplementary Figure 1E), all of which have high endogenous *TMEM106B* levels (Supplementary Figure 1D). *TMEM106B* knockdown cells demonstrated a significant reduction in migration and invasion in vitro (Fig. 2c and Supplementary Figure 1F). Further, knockdown of *TMEM106B* in 344SQ cells strongly inhibited the TGFβ-mediated invasive structures, when grown in a 3D matrix (matrigel) (Fig. 2d). To test whether downregulation of *TMEM106B* expression altered the in vivo metastatic potencies, the syngeneic mouse lung cancer cell lines (344SQ and 344LN) with *TMEM106B* knockdown or Control (Scramble) were injected subcutaneously in syngeneic mice. Both groups formed comparable sized primary tumors by 7–8 weeks (Fig. 2e and Supplementary Figure 1B), but the *TMEM106B* knockdown cells exhibited robust repression of lung metastases when compared to the control scrambled shRNA cells. These observations were evident from scoring the macroscopic metastatic lung nodules and confirmed by H&E-staining of lung sections (Fig. 2f, g and Supplementary Figure 1C). These data indicate *TMEM106B* is required for maintaining full cell migration, invasion and metastasis phenotypes exhibited by these murine and human cell models.

***TMEM106B* induces synthesis of enlarged functional lysosomes**. *TMEM106B* is a single pass, type-II transmembrane protein reported to localize on lysosomes. We first sought to determine the localization and functional implication of *TMEM106B* expression by establishing both murine (344SQ, 393P, and LLC-JSP) and human (H157 and HCC827) cell lines with inducible expression of full-length, green fluorescent protein (GFP)-tagged *TMEM106B*. We observed that irrespective of the cell type, within 24–48 h of *TMEM106B* induction there was robust expression of the GFP-tagged *TMEM106B* (Supplementary Figure 2A), which was associated with formation of enlarged membrane bound vesicles in comparison to GFP-expressing control cells, which has also been reported in other cell types upon ectopic expression of *TMEM106B*[14,37–39], (Fig. 3a1–2, b1–2 and c1–2 and Supplementary Figures 2B1-2, 2D). Strikingly, the membranes were GFP positive, consistent with the membrane localization of the fusion protein. Staining live cells with Lysotracker (acidic lysosome marker) and imaging demonstrated that these vesicular lumens exhibited positive staining, confirming them as endolysosomes (Fig. 3a3–5, b3–5 and c3–5 and Supplementary Figure 2B3–5, D2–3). The lysotracker staining was quantified by flow cytometry or by measuring fluorescence intensity, to confirm that the TMEM106B-expressing cells display significant more staining (Supplementary Figure 2C and 2D4), as compared to GFP-expressing control cells.

Functional lysosomes contain bioactive hydrolases and proteases, among which cathepsins comprise one of the most abundant family of proteases. To ascertain the functionality of these enlarged lysosomal vesicles formed as a result of *TMEM106B* expression, we wanted to determine the presence of bioactive cathepsins in these organelles. For this we performed a fluorescence based assay (Magic Red) to monitor and quantitate intracellular cathepsin B activity in real time in live cells upon *TMEM106B* induction. We observed that in each of the different

human and murine tumor cell models, upon *TMEM106B* induction, the enlarged lysosomes are loaded with active cathepsin B as evident from the enhanced red fluorescence in the lysosomal vesicles (Fig. 4a–c and Supplementary Figure 3D) and significant increase in florescence intensity (Fig. 4d and Supplementary Figure 3E). A similar increase in activity was observed and quantified for cathepsin K in the *TMEM106B*-induced cells compared to the GFP controls (Supplementary Figures 3A, B, F, G). Additionally, we also observed increased levels of active cathepsin B and D proteins in the human cells induced for *TMEM106B* expression and reduced levels upon *TMEM106B* knockdown (Supplementary Figure 3C).

**TMEM106B drives *TFEB*-mediated regulation of CLEAR genes**. According to recent findings[40,41], *TFEB* is the master regulator of lysosome synthesis and function, where it transcriptionally regulates expression of lysosomal genes of the CLEAR pathway[41]. Since *TMEM106B* has been shown to regulate *TFEB* activity in neuronal cells, we wanted to determine whether *TMEM106B* similarly regulates *TFEB* in a manner that is required for *TMEM106B*-driven phenotypic changes in lung cancer cells. First, we wanted to determine if *TMEM106B* could regulate localization of *TFEB*. Co-expression of Mcherry tagged *TFEB* in cells induced for GFP-*TMEM106B* expression produced enhanced nuclear localization of *TFEB* compared to cells co-expressing GFP only, where *TFEB* was mostly cytoplasmic (Fig. 5a). This observation was further validated by cell fractionation and western blot analyses, which showed endogenous *TFEB* was elevated in the nuclear fractions of *TMEM106B* over-expressing cells compared to cells expressing GFP only (Fig. 5b and Supplementary Figure 4A) and in cells where *TFEB* was exogenously overexpressed with *TMEM106B* versus GFP only (Supplementary Figure 4B). Lamin and tubulin blotting were included as fractionation controls for nuclear and cytoplasmic fractions, respectively. It was also interesting to observe that out of the top five hits for metastasis drivers (as tested in Fig. 1a) only the cells overexpressing *TMEM106B* showed a prominent nuclear/cytoplasmic enrichment of *TFEB*, indicating specific regulation of *TFEB* by *TMEM106B* (Fig. 5c). We also wanted to determine whether *TMEM106B* was critically necessary for nuclear translocation of *TFEB*. For this we treated either scramble control or *TMEM106B* knockdown cells with increasing concentrations of Torin 1, which is known to induce *TFEB* nuclear translocation, by inhibition of *mTORC1*[42]. We observed that both in the control and *TMEM106B* knockdown cells, *TFEB* could translocate to the nucleus upon Torin 1 treatment (Fig. 5d). Our results indicate that though expression of *TMEM106B* could drive nuclear translocation of *TFEB*, it is not critically necessary for the process.

We determined that *TMEM106B* expression resulted in upregulation of almost all of the CLEAR lysosomal genes we examined through RT-qPCR analysis of 393P and 344SQ cells (Fig. 4c and Supplementary Figure 5E), whereas these genes were repressed upon shRNA-mediated *TMEM106B* knockdown in metastatic lung cancer cells (Fig. 5d and Supplementary Figure 1A). Further, when we analyzed gene expression

**Fig. 2** TMEM106B knockdown suppresses invasion and metastasis. **a** qPCR analysis for TMEM106B expression in 344SQ cells with stable expression of shRNA targeting TMEM106B versus a scramble control. **b** Western blot analysis for TMEM106B expression in 344SQ cells with stable expression of shRNA targeting TMEM106B. **c** The TMEM106B knockdown cells demonstrate reduced ability to migrate and invade compared to scramble controls. **d** The knockdown cells suppress formation of invasive structures in a 3D matrix upon treatment with TGFβ. **e** The knockdown cells were injected subcutaneously into syngeneic hosts and observed for primary tumor growth and **f** formation of metastatic nodules in lungs. **g** Representative lungs and their respective H&E sections are shown. Lungs from mice injected with TMEM106B knockdown cells show significantly reduced incidence of lung metastasis. All asterisks indicate statistical significance by $t$ test ($n \geq 3$, $^*p \leq 0.05$)

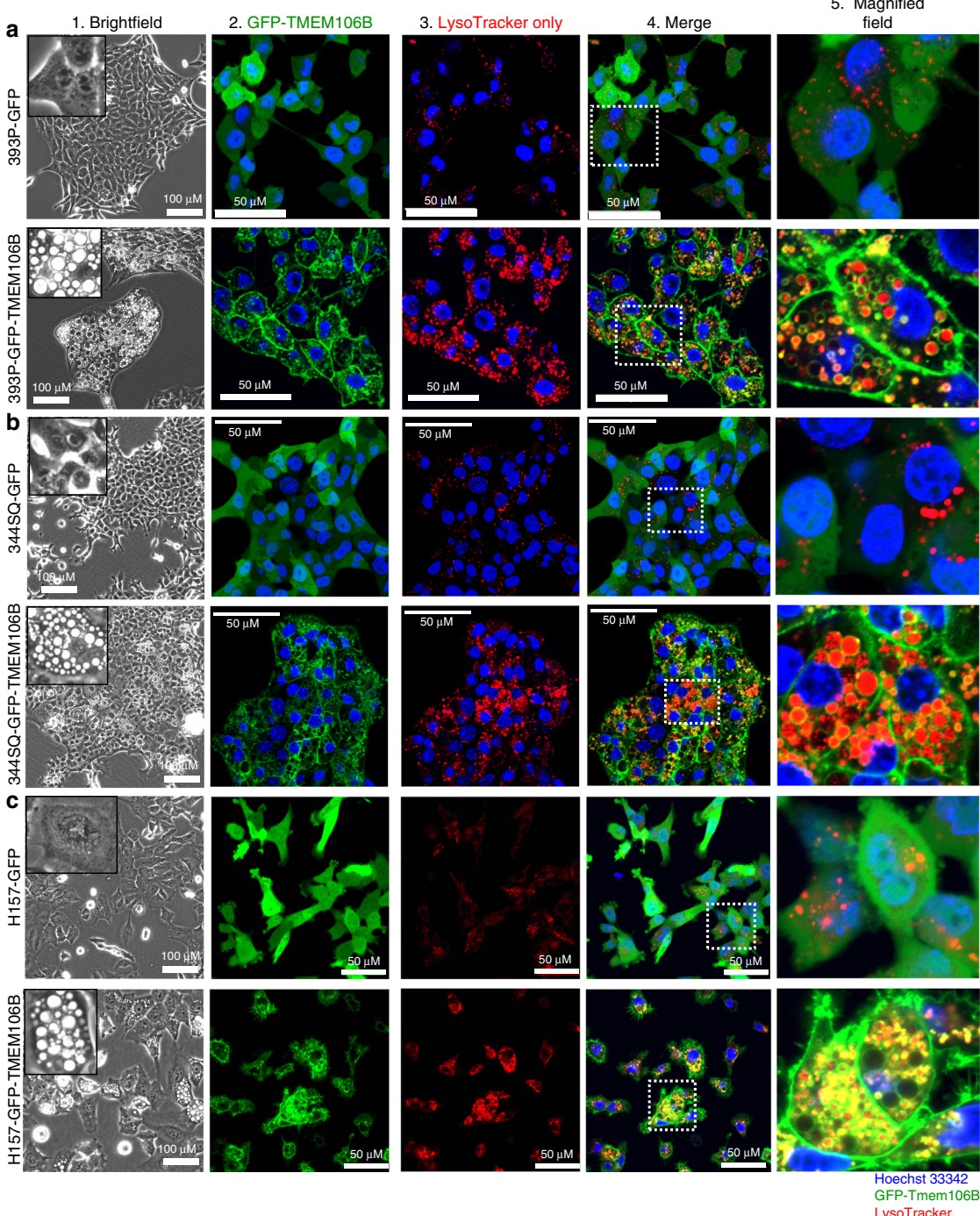

**Fig. 3** TMEM106B induces synthesis of increased number and size of lysosomes. **a–c** Human or mouse lung cancer cells induced to express either GFP as control, or GFP-tagged TMEM106B were imaged as indicated. (1) Cells imaged in bright field show enlarged vesicular structures formed upon expression of TMEM106B. (2–5) Cells were incubated with LysoTracker stain and counter stained with Hoechst33342, imaged live in real time as indicated. TMEM106B-expressing cells form enlarged lysosomes that stain positive for LysoTracker. Magnified fields exhibit the stained vesicles

correlations from an extensive data set of more than 1000 human lung cancer samples, *TMEM106B* expression demonstrated significant direct correlation with expression of more than 60% of the CLEAR genes that were included in the analysis, including cathepsins B and D (Fig. 5e). Interestingly we did not observe any significant change in mRNA levels of *TFEB* expression, in cells or clinical samples with increased *TMEM106B* expression (Fig. 5e and Supplementary Figure 5H). To directly test the role of *TFEB* downstream of *TMEM106B* function, we transiently depleted *TFEB* expression by siRNA in control cells (Vec+) or cells expressing *TMEM106B* (Fig. 5f). We observed that the upregulation of the CLEAR genes upon *TMEM106B* induction was completely and significantly reversed upon knockdown of *TFEB* (Fig. 5g). When we determined the activity of intracellular cathepsin B and K in these cells by Magic Red activity assay, we observed that the increase in the cathepsin activity upon *TMEM106B* induction was completely and significantly reversed with *TFEB* knockdown (Fig. 5h). Further we wanted to determine whether *TMEM106B* was a direct transcriptional target of *TFEB*. For this we either transiently overexpressed or knocked down

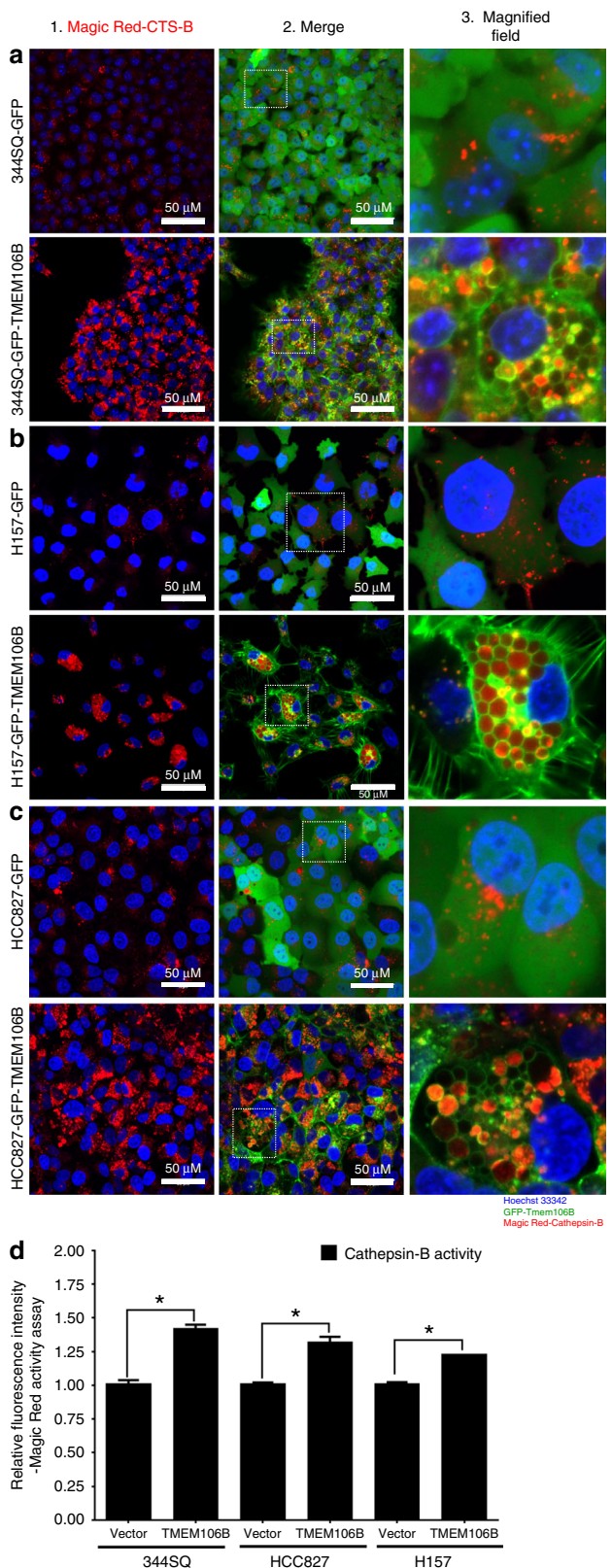

**Fig. 4** TMEM106B-induced lysosomes are loaded with catalytically functional cathepsins. **a–c** Cells expressing GFP only or GFP-tagged TMEM106B were incubated with Magic Red Cathepsin B reagent to stain for activity of cathepsin B and counter stained with Hoechst33342. Cell were imaged live in real time as indicated. **d** Quantification of fluorescence intensity corresponding to cathepsin B activity as analyzed by Magic Red Cathepsin B activity assay in different cells. All asterisks indicate statistical significance by $t$ test ($n \geq 3$, $^{*}p \leq 0.05$)

*TFEB* and tested whether these changes affected *TMEM106B* expression. We observed that in either scenario there was no significant change in *TMEM106B*, at the mRNA (Supplementary Figure 4F) or protein (Supplementary Figure 4G) level, indicating that *TMEM106B* expression was not regulated by *TFEB*. Taken altogether, these results suggest that *TMEM106B* expression in lung cancer cells results in increased lysosome size and enhanced activity of *TFEB*, thereby modulating expression of the CLEAR genes.

**TMEM106B induces lysosomal exocytosis**. Increased production of lysosomal cathepsin B and L could act as signals for increased lysosome trafficking to the plasma membrane[22,25,26], where lysosomes are fused to each other before fusing with the plasma membrane causing lysosomal exocytosis, usually as a result of a calcium flux[39]. This final fusion of the lysosomes to the plasma membrane results in the transfer of the luminal sialylated lysosomal membrane protein *LAMP1* to the extracellular side of the plasma membrane and also releases the complete lysosomal content into the extracellular matrix. To determine whether lung cancer cells with induced *TMEM106B* expression show increased lysosomal exocytosis, we co-expressed RFP-tagged *LAMP1* in the cells expressing GFP-tagged *TMEM106B* or GFP as control. The cells expressing both of the fluorescently tagged proteins demonstrated extensive co-localization of *TMEM106B* and *LAMP1* in the enlarged lysosomal vesicles (Fig. 6a and insets). Interestingly, these cells also exhibited extensive localization of both *LAMP1* and *TMEM106B* at the plasma membrane, indicating that these cells are actively undergoing lysosomal exocytosis. To further validate this observation, we undertook a biochemical approach and performed cell fractionation assay using cells expressing GFP vector only or GFP-tagged *TMEM106B*. As expected, we observed higher expression of *LAMP1* in the whole cell extracts of cells induced with TMEM106B expression. The TMEM106B overexpressing cells also exhibited significantly higher abundance of *LAMP1* in the plasma membrane fractions compared to the vector control cells, indicating elevated lysosomal trafficking to the membrane, with associated exocytosis (Fig. 6b). To confirm that cells induced for *TMEM106B* expression undergo active lysosomal exocytosis, we performed immune fluorescence surface staining for *LAMP1* to detect its presence on the cell membrane. This was performed by first fixing the cells using paraformaldehyde followed by immune staining with a phycoerythrin (PE)-conjugated *LAMP1* primary antibody, without permeabilization, to specifically stain the *LAMP1* present on the cell surface due to fusion with the lysosomal membrane caused during exocytosis. We observed elevated surface staining of *LAMP1* in *TMEM106B* induced cells compared to control (Fig. 6c). This was also quantified by flow cytometric analysis of stained cells (Supplementary Figure 4H).

We observed that in the conditioned media of *TMEM106B*-expressing cells compared to the conditioned media of GFP vector control cells, there was higher abundance of secreted lysosomal proteins (Fig. 6d) and consistently higher activity of cathepsins B and D, as measured by fluorimetric activity assays (Fig. 6e, f). We also observed that in comparison to the conditioned media collected from cells expressing GFP only, the conditioned media from cells induced for *TMEM106B* expression was able to better stimulate parental non-invasive 393P cells to invade in Boyden chamber invasion assays (Fig. 6g). We also determined that there was no toxicity induced by *TMEM106B* induction as conditioned media from both control and *TMEM106B* cells demonstrated comparable LDH activity (Supplementary Figure 4I). These results suggest that *TMEM106B* expression in lung cancer cells leads to elevated lysosome

production, lysosomal exocytosis, and secretion of lysosomal cathepsins into the ECM that may enhance the invasion, migration and metastasis of the tumor cells.

**TMEM106B drives metastasis by release of lysosomal cathepsins.** Elevated cathepsin activity and extracellular secretion play

a crucial role in ECM degradation and dissolution of cell–cell adhesion molecules, which are contributory toward cancer cell invasion and metastasis[24,31–35]. We wanted to determine whether elevated cathepsin synthesis and release by lysosomal exocytosis was the major mechanism by which *TMEM106B* enhances invasion and metastasis. For this we leveraged the fact that

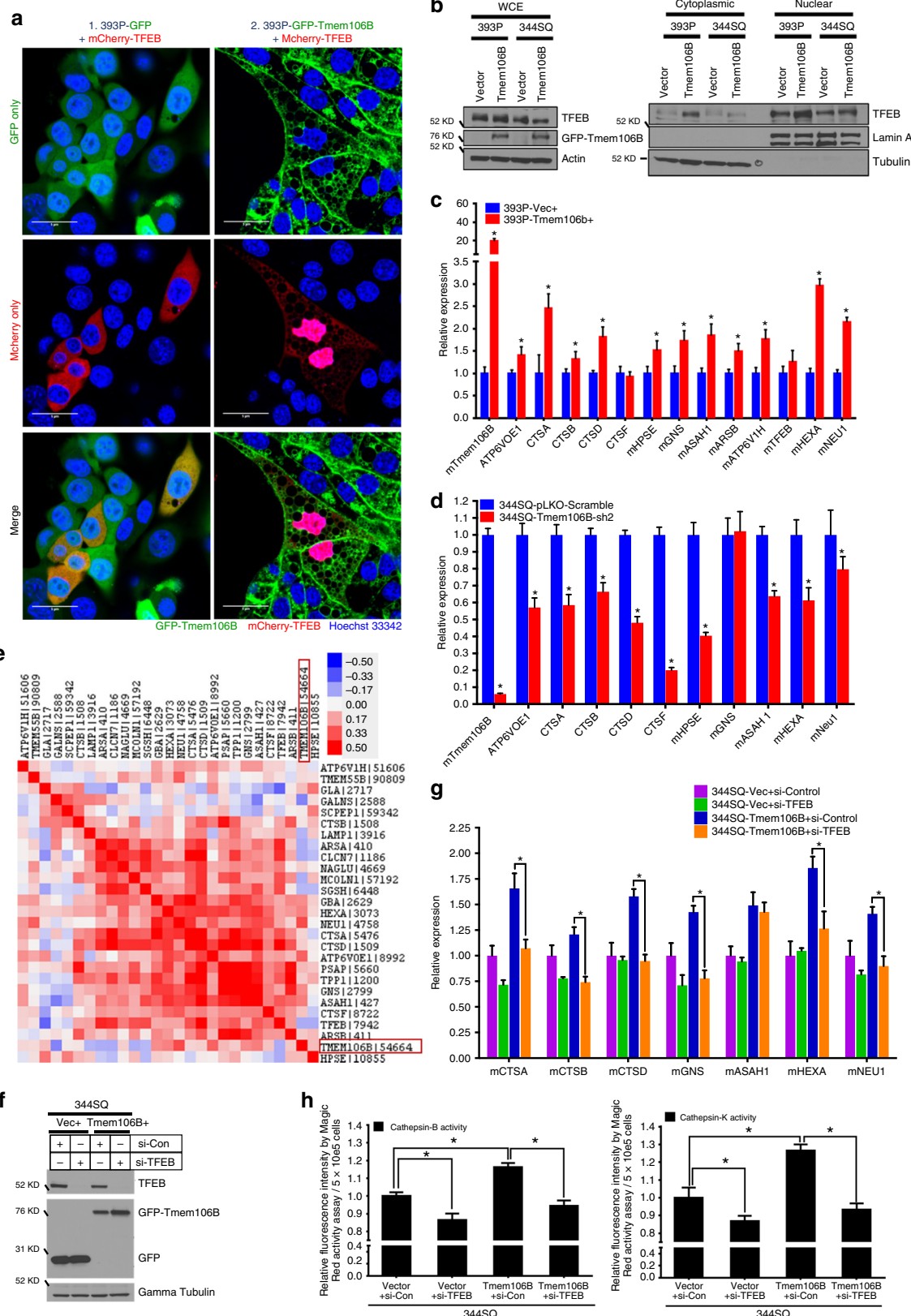

lysosomes behave as $Ca^{2+}$-regulated exocytic vesicles[39] to release the lysosomal content from cells expressing GFP control or *TMEM106B*, using a brief treatment with ionomycin in phosphate buffered saline (PBS) containing $Ca^{2+}$. We observed that upon ionomycin-mediated lysosomal exocytosis, the conditioned PBS for the TMEM106B-expressing cells demonstrated significantly higher levels of activity for cathepsin B, D, H, and K, compared to ionomycin-treated conditioned PBS from GFP-expressing control cells (Vec), or DMSO-treated controls. The activities of the cathepsins were measured by specific fluorimetric activity assays, which were inhibited by the inclusion of specific inhibitors (Fig. 7a, b and Supplementary Figure 5A, B). We also noted that the ionomycin-mediated release of the lysosomal contents was completely dependent on the presence of external $Ca^{2+}$, without which there was no difference in activity of cathepsin B in the ionomycin-treated conditioned PBS from either GFP or TMEM106B-expressing cells (Supplementary Figure 5C, D). Next, we asked whether the active cathepsins released upon lysosomal exocytosis were essential to induce an invasive phenotype. To test this, we performed Boyden chamber invasion assays with noninvasive 393P cells suspended in either ionomycin-treated conditioned PBS from cells expressing GFP (Vec) or *TMEM106B*. We observed that the 393P cells were significantly more invasive when incubated with conditioned PBS of *TMEM106B* cells compared to GFP (Vec) cells and this invasive phenotype was completely abrogated when the assay was performed in the presence of a protease inhibitor cocktail (Fig. 7c) or in the presence of an irreversible cysteine protease inhibitor that specifically inhibits cathepsins B, H, and K (Fig. 7d). The results indicate that *TMEM106B* induces the production of enlarged lysosomal vesicles laden with bioactive cathepsins, which undergo $Ca^{2+}$-dependent lysosomal exocytosis that enhances the invasion and migration of the tumor cells. Next, we demonstrated that cells with inducible expression of *TMEM106B* could form larger primary tumors with significantly increased metastasis in vivo, when implanted into syngeneic mice (Supplementary Figure 6C).

To further ascertain whether elevated cathepsin synthesis and release was the major mechanism by which *TMEM106B* induces metastasis in vivo, we performed a therapeutic intervention study. The murine 344SQ cells with inducible expression of Vector (GFP only) or *TMEM106B* were implanted into syngeneic mice. These mice were fed with doxycycline-containing feed. Half of the mice from either vector or *TMEM106B*-cell injected cohorts were treated with the irreversible cysteine protease inhibitor E64D (Aloxistatin, 25 mg/kg body weight, oral daily), and the remaining half were dosed with the vehicle as control (Supplementary Figure 6A). After 4 weeks of treatment we observed that the primary tumors of the *TMEM106B*-cell injected mice had about threefold higher expression of *TMEM106B* (Fig. 8a and Supplementary Figure 6B). We noted that all the tested cathepsins demonstrated significant increased activity in the TMEM1066B primary tumors compared to the vector control tumors, within the

vehicle treated cohorts. This strengthened our observation that *TMEM106B* expression induced increased production of active cathepsins. Further, we observed that in the E64D-treated cohorts, the activity of cathepsins B, K, and H were significantly suppressed in the primary tumors and completely inhibited in the liver (Fig. 8b–d), when compared to the vehicle treated cohort samples, indicating effective systemic delivery of the drug. Cathepsin D, which is an aspartyl protease, was unaffected by E64D (cysteine protease inhibitor) treatment, suggesting specificity of the inhibitor (Fig. 8e). Finally, we observed that there was no significant difference in primary tumor sizes (Fig. 8f) between the E64D-treated and vehicle treated mice. However, the enhanced metastasis that was observed in the vehicle treated *TMEM106B*-induced cohort, was significantly reverted in the E64D-treated *TMEM106B*-induced mice (Fig. 8g, h). These results demonstrate that the elevated synthesis and release of active lysosomal cathepsins upon *TMEM106B* expression, could create an invasive microenvironment conducive for metastatic spread of cancer cells and could be therapeutically targeted using a cathepsin inhibitor.

**TMEM106B expression is elevated in human lung adenocarcinomas.** Our data outline a novel mechanism for *TMEM106B*-mediated metastasis in lung cancer. Next, we wanted to assess the clinical relevance of these findings so as to determine the status of *TMEM106B* expression in human lung adenocarcinoma data sets and whether its expression profile could be associated as a prognostic indicator of disease outcome. For this we queried the TCGA data sets for lung adenocarcinoma (TCGA-Provisional as queried using http://www.cbioportal.org/[43,44] and accessed through this link: http://bit.ly/2q1FssL), which have been fully annotated for gene expression and contain associated follow-up data on patient outcome. According to this TCGA lung adenocarcinoma data set, containing 517 samples, 19% of the samples ($n = 98$) show either gene amplification or upregulation of the mRNA expression for *TMEM106B* (Supplementary Figure 5E, F). The patients with amplification or increased *TMEM106B* expression in their primary lung tumors had more frequent relapses with a significantly (log rank $p = 0.0059$) worsened disease-free survival and decrease in median survival by 19 months, from 41.23 to 22.54 months (queried using http://www.cbioportal.org/[43,44]) (Fig. 9a). The patients with elevated *TMEM106B* levels also had significantly (log rank $p = 0.0041$) worse overall survival, with a decrease in median survival by 16 months, from 53.29 to 37.68 months (Fig. 9b). Interestingly, *TMEM106B* gain or amplification status in human nonsmall cell lung cancer (NSCLC) patients is not significantly associated with the KRAS mutational status, indicating that *TMEM106B* is an independent prognostic driver of lung cancer metastasis (Supplementary Figure 5G). We also wanted to determine whether *TFEB* expression status in lung adenocarcinoma could be associated as a prognostic indicator of disease outcome. In the TCGA lung adenocarcinoma data set

---

**Fig. 5** TMEM106B drives nuclear translocation of TFEB and regulates TFEB-mediated expression of lysosomal genes. **a** 393P cells with inducible expression of TMEM106B or GFP alone were transfected with Mcherry-TFEB construct. After induction for 24–48 h, cells were counter stained with Hoechst33342. Live cells were imaged in real time. Indicated scale bar = 5 μm. **b** Western blot analysis upon subcellular fractionation of cells with induction of GFP-TMEM106B or GFP alone as control. **c**, **d** qPCR analysis for expression of lysosomal genes upon expression or knockdown of TMEM106B expression. **e** Heatmap of intergene expression correlation across 1016 lung cancers from TCGA[49]. Color bar represents Pearson's $r$ value (based on log-transformed values); red is positive correlation. **f** Western blot analysis after transfection of either control siRNA or siRNA against TFEB in cells expressing either GFP alone or GFP-TMEM106B. **g** qPCR analysis for expression of lysosomal genes upon induction of GFP alone as control (Vec) or TMEM106B expression, after transfection of either control siRNA or siRNA against TFEB in cells expressing either GFP or GFP-TMEM106B. **h** Relative cathepsin B or cathepsin K activity as measured by quantification of fluorescence intensity after staining with Magic Red Cathepsin B or K reagent to assess for activity of cathepsin B or K, in cells as described in **g**. All asterisks indicate statistical significance by $t$ test ($n \geq 3$, $^*p \leq 0.05$)

only 4% of cases show amplification of *TFEB* (data not shown). We also observed that patients with elevated *TFEB* had no significant difference in disease outcome when compared to unaltered control patient group (Supplementary Figure 5I). The data therefore suggest that *TMEM106B* could-potentially be utilized as a prognostic marker and indicator of poor disease outcome and importantly to be developed

for intervention strategies to target lung adenocarcinoma growth and metastasis.

Based on our findings we propose a model by which elevated levels of *TMEM106B* in primary lung cancers lead to an increased nuclear translocation of *TFEB*. This results in a *TFEB*-dependent upregulation of lysosome genes of the CLEAR pathway, which are loaded in enlarged vesicular lysosomes. Upon intracellular

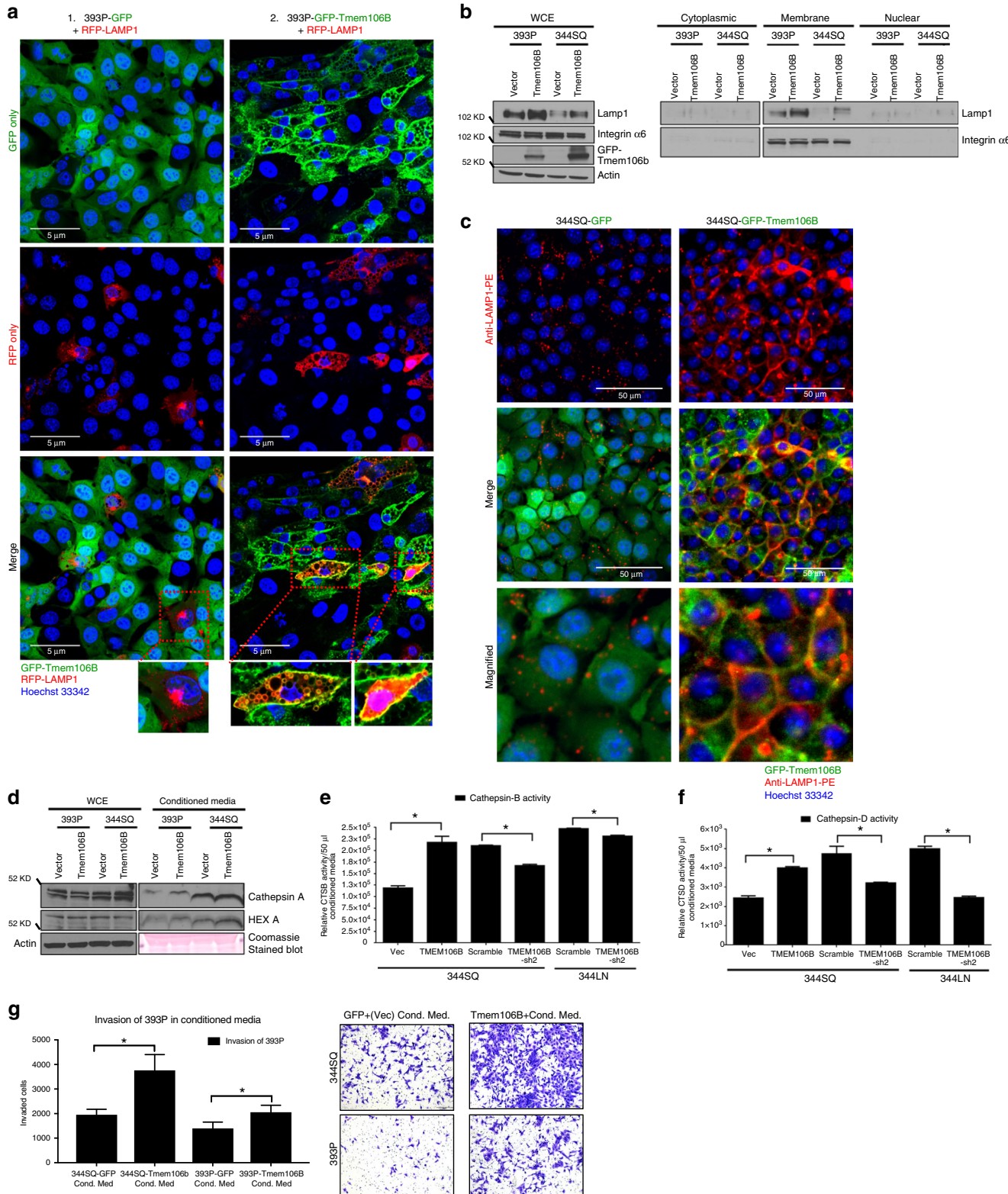

calcium influx, these lysosomal vesicles are secreted by lysosomal exocytosis, releasing a host of lysosomal proteases, including a variety of cathepsins into the extracellular matrix. The release of these cathepsins and other hydrolases increases tumor cell invasion and metastasis (Fig. 9c).

## Discussion

The molecular mechanisms underlying lung cancer metastasis are still largely unclear. To identify novel genes involved in this complex multistep process we performed an in vivo gain-of-function screen for drivers of lung cancer metastasis[5]. Importantly we implemented an approach using a pooled viral infection strategy that takes into account the importance of genetic context and tumor microenvironment in selection of single and combinatorial drivers of metastasis in an immune-competent environment. The nature of the screen ensured stringency in the selection of candidate drivers that can be further investigated with high confidence as functional mediators of lung cancer metastasis. Our results established TMEM106B as an efficient and novel driver gene for metastasis. Although TMEM106B has a high frequency of gain or amplification in NSCLC samples, its role as a driver of tumorigenesis or metastasis has not been reported. This is the first report of a mechanistic link demonstrating how TMEM106B could drive lung cancer metastasis by deregulating lysosome function.

Dysfunctional lysosomes have been associated with several pathological disorders including inborn lysosome storage disorders, neuro-degenerative disorders like Alzheimer's, Parkinson's, Huntington's, and frontotemporal disorder[45]. Lysosome functions have been reported to be altered during cancer progression, where lysosomal volume, composition, and subcellular localization undergo abnormal changes[21]. Lysosomal cathepsins are highly up-regulated, and mislocalization of lysosomes during neoplastic progression results in their elevated secretion[24]. Several cathepsins, such as cysteine cathepsins B and L and the aspartate cathepsin D among others, have been implicated in the progression of different cancer types. High expression levels of these cathepsins are frequently associated with metastasis and poor prognosis[22]. Lysosomal biogenesis is tightly controlled by either the metabolic status of the cell, or signaling through the lysosomal membrane bound molecules (mammalian target of rapamycin [mTOR] and TFEB), which has been shown to regulate not only lysosomal function, but also overall cellular metabolism. According to recent findings[40,41], one of the master regulators of lysosome synthesis and function is the transcription factor TFEB, which transcriptionally regulates the biogenesis and function of active lysosomes and late endosomes. Most of the lysosomal genes contain a conserved palindromic motif named the CLEAR element in their promoter, which facilitates direct binding by TFEB and subsequent transcriptional regulation[41]. Accordingly, ectopic expression of TFEB results in an elevated number and increased luminal content of lysosomes, thus enhancing lysosomal catabolic activity[41]. We demonstrate that TMEM106B acts upstream of TFEB, with increased TMEM106B

expression causing nuclear translocation of TFEB and resulting in alteration of lysosomal content and function. Interestingly, this role of TMEM106B appears to be operating independently from other stimuli that regulate TFEB nuclear translocation. These results suggest that TMEM106B expression in lung cancer cells results in increased lysosome size and enhanced activity, describing a new role for TMEM106B in regulating lysosome function in lung cancer metastasis. We also wanted to address whether TMEM106B could be directly regulated by TFEB. To test this, we first performed a motif search on the TMEM106B promoter (using JASPAR online tool: http://jaspar.genereg.net/), but were unable to identify any high scoring known TFEB binding motif on the TMEM106B promoter. This is consistent with published literature where TMEM106B was not identified as a TFEB transcriptional target[46]. Our work lays the foundation for future work to address the molecular mechanisms underlying the posttranslational changes of TFEB upon TMEM106B expression that result in its nuclear translocation.

Lysosomal exocytosis occurs due to a variety of stimuli, which are responsible for the onset of different pathological states. In cancer, the trafficking and fusion of lysosomes with the plasma membrane contribute to cell invasion and migration, making this phenomenon of crucial importance for our metastasis research[22–24]. Elevated cathepsin activity and extracellular secretion/release by lysosomal exocytosis plays a crucial role in tumor progression and metastasis in several cancers through various mechanisms[24,47]. This is the first report that elaborates the novel connection between TMEM106B overexpression in lung cancer cells and elevated lysosome production. These lysosomes are actively secreted by lysosomal exocytosis in a calcium-dependent manner, which leads to increased secretion of lysosomal cathepsins in the ECM, facilitating enhanced invasion, migration, and metastasis. This study also invokes the need for further analysis to understand the exact molecular mechanism of anterograde trafficking of the enlarged lysosomes to the plasma membrane. We acknowledge the fact that cancer cells with elevated TMEM106B expression produce very large amounts of intracellular active cathepsins. As a result, there could be additional mechanisms that are co-activated and partially induce direct secretion of a fraction of the proteases from the trans-Golgi network to the plasma membrane, without being properly sorted into the lysosomes. This possibility needs further investigation that is beyond the scope of the current manuscript.

Our data here identify a novel function of TMEM106B, where its expression in lung cancer cells stimulates the production of enlarged lysosomes and increased synthesis of lysosomal hydrolases that are packaged into the lysosomes. These lysosomes are actively secreted under conditions of calcium flux, releasing their cargo of proteases into the extracellular matrix and producing a hyper-invasive microenvironment. We also demonstrate that systemically impeding the activity of the proteolytic enzymes with a pharmacologic inhibitor could revert the metastatic phenotype caused by elevated expression of TMEM106B. Therefore, this experimental evidence suggests that targeting TMEM106B directly or its

**Fig. 6** TMEM106B induces lysosomal exocytosis. **a** 393P cells with inducible expression of GFP-TMEM106B or GFP alone were transfected with an RFP-LAMP1 construct. After induction for 24–48 h, cells were counter stained with Hoechst33342. Live cells were imaged in real time. Indicated scale bar = 5 μm. **b** Western blot analysis upon subcellular fractionation of cells with induction of GFP-TMEM106B or GFP control. **c** Immunofluorescence staining for endogenous LAMP1 on outer plasma membrane of cells. 344SQ cells induced for either GFP only as control or GFP-tagged TMEM106B were first fixed using paraformaldehyde followed by immune staining using a phycoerythrin (PE)-conjugated LAMP1 primary antibody, without permeabilization, to specifically stain the endogenous LAMP1 present on the cell surface. **d** Western blot analysis for secreted lysosomal proteins in the conditioned media of cells expressing GFP-TMEM106B or GFP. **e** Cathepsin B activity measured using specific activity assay, in 50 μl of conditioned media from cells with either TMEM106B induction or knockdown, as indicated. **f** Cathepsin D activity measured using specific activity assay, in 50 μl of conditioned media from cells with induction or knockdown of TMEM106B expression, as indicated. **g** Invasion of non-invasive 393P cells when suspended in conditioned media from cells induced for GFP control or TMEM106B. All asterisks indicate statistical significance by t test ($n \geq 3$, $^*p \leq 0.05$)

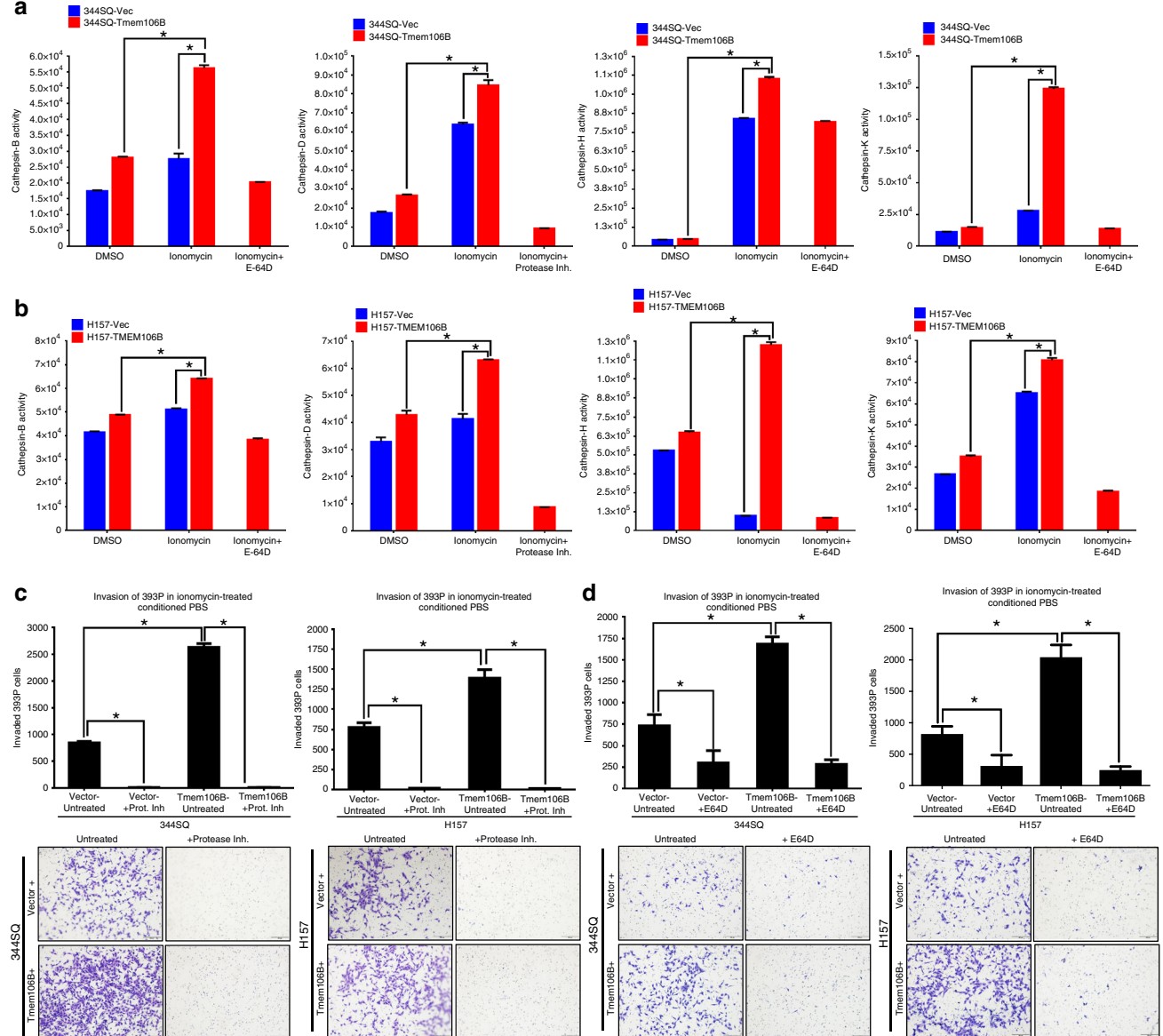

**Fig. 7** TMEM106B drives invasion by inducing elevated production and release of cathepsins. **a** Activity of cathepsins were assessed using specific activity assays in conditioned phosphate buffered saline (conditioned PBS) where lysosomal contents were released by treating mouse, or **b** human cells expressing either GFP only or TMEM106B, briefly with ionomycin or DMSO (as control) in the presence of 2 mM $Ca^{2+}$. Assays were also performed in the presence of specific inhibitors, as indicated. **c** Invasion assays performed with non-invasive 393P cells, which were incubated with conditioned PBS of human or mouse cells induced for either GFP (Vec) as control or TMEM106B, in the presence of either DMSO as control or a protease inhibitor cocktail. **d** Invasion assays performed with noninvasive 393P cells, which were incubated with conditioned PBS of human or mouse cells induced for either GFP (Vec) or TMEM106B, in the presence of either DMSO or the irreversible cysteine protease inhibitor (E64D), which specifically inhibits cathepsins B, H, and K

downstream functions could be developed into a novel therapeutic intervention strategy to prevent metastatic lung cancers.

## Methods

**Plasmids and reagents**. Mouse *TMEM106B* cDNA was amplified by RT-PCR as a MfeI–XhoI fragment (primers as in Supplementary Table 1) and cloned as a GFP fusion construct in the dox inducible pTRIPZ-GFP vector where the miR-30~RFP cassette was replaced with the GFP cDNA[48]. Mouse *TFEB* cDNA was amplified by RT-PCR as a HindIII–BamHI fragment and cloned as a Mcherry fusion construct (primers as in Supplementary Table 1) in pMcherry-C3 plasmid. RFP-*LAMP1* expressing plasmid construct was purchased from Addgene. Mouse *TMEM106B* shRNA construct was purchased from Open-Biosystems/GE-Dharmacon and Scramble control vectors were purchased from Thermo-Scientific. Human *TMEM106B* shRNAs (TRCN0000123314 and TRCN0000130047), and Scramble control vectors were purchased from Thermo-Scientific. siRNA for *TFEB*

(SMARTpool:ON-TARGETplus) was purchased from Dharmacon/GE-Healthcare (Lafayette, CO).

**Cell culture and transfections**. All cell lines were maintained in RPMI-1640 supplemented with 10% fetal bovine serum. DNA transfections were performed with Lipofectamine-2000 or Lipofectamine-LTX reagents (Life technologies, Grand Island, NY), miRNA precursors were transfected at 50 nM final concentration for 96 h using Lipofectamin-2000 (Life technologies) and siRNAs were transfected at 25 nM final concentration using RNAi-Max (Life technologies, Grand Island, NY) reagents as per manufacturer's protocols. All mouse cell lines were generated in our lab and authenticated[7]. NCI-H157 cells were procured from ATCC and tested negative for mycoplasma.

**qPCR and western blot analysis**. Total RNA was isolated and RT-PCR was performed using specific primers (Supplementary Table 1) and SYBR® Green PCR Master Mix (Life technologies). SYBR green qPCR analyses were normalized to

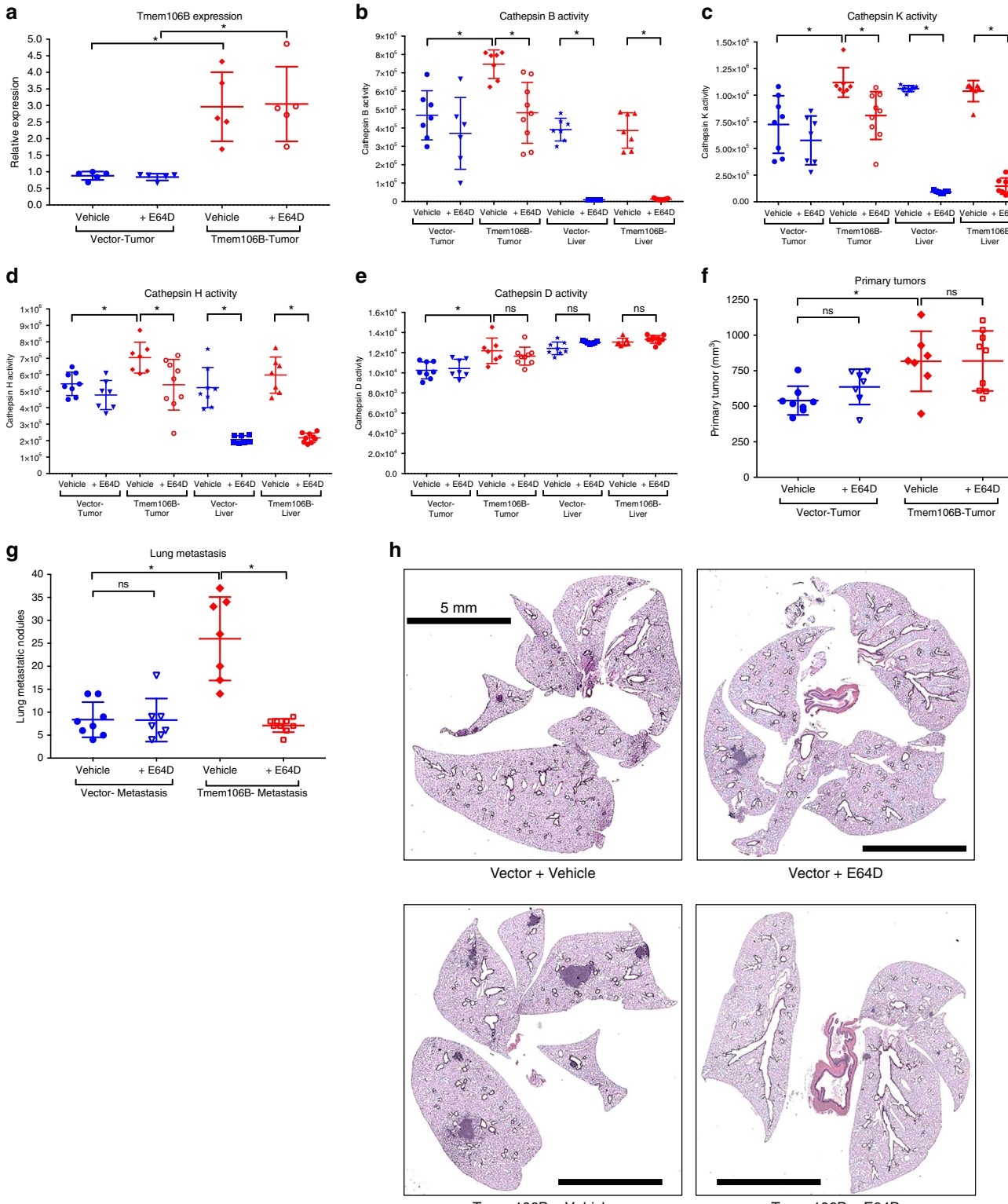

**Fig. 8** Inhibition of cathepsins in vivo results in abrogation of *TMEM106B*-mediated increase in lung metastasis. The Vector alone or *TMEM106B*-induced cells were injected subcutaneously into syngeneic hosts and treated daily with either Vehicle control (Placebo) or with 25 mg/kg of E64D. **a** qPCR analysis for *TMEM106B* expression in primary tumor tissue from mice of different cohorts as indicated. **b–e** Activity of indicated cathepsins were assessed using specific activity assays in tissue lysates from either primary tumors or liver in the different cohorts as indicated. **f** Observed primary tumor growth and **g** formation of metastatic nodules in lungs. **h** Representative H&E-stained lungs sections are shown from different cohorts as indicated. Lungs from mice injected with *TMEM106B*-induced cells show significantly more incidence of lung metastasis which are abrogated upon treatment with E64D. All asterisks indicate statistical significance by *t* test (*n* ≥ 3, *p* ≤ 0.05)

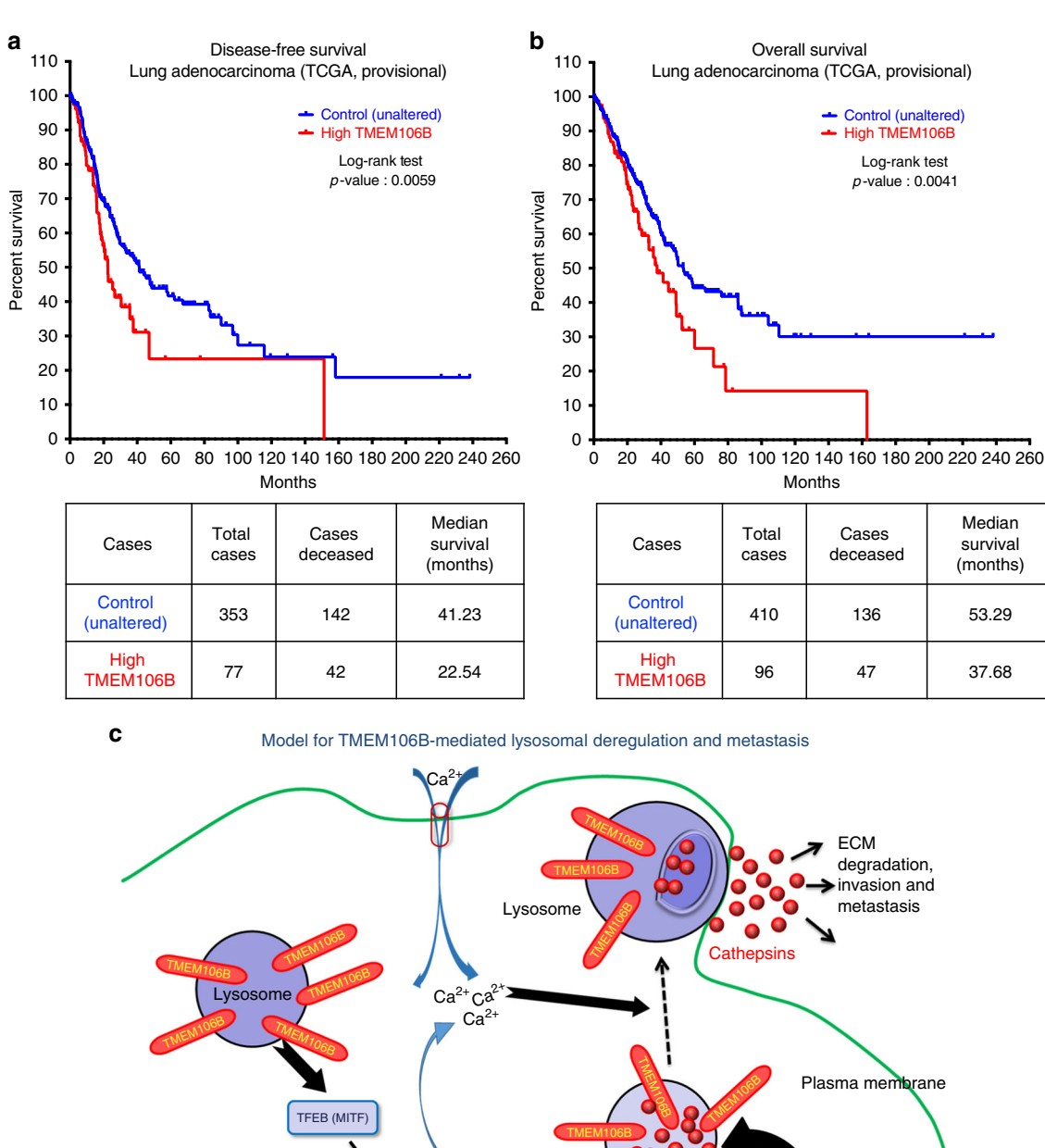

**Fig. 9** *TMEM106B* expression is elevated in human lung adenocarcinomas and predicts for poor prognosis. **a** Overall survival and **b** disease-free survival plots of lung adenocarcinoma patents with elevated expression of *TMEM106B* in TCGA (provisional, *n* = 517) data set. Graphical representation of respective case numbers and median survivals are shown below. **c** Schematic of model for *TMEM106B*-mediated deregulation of lysosomal function in cancer cells

expression of L32 (60S ribosomal gene). Taq-man assays (Life technologies) for miRNA qPCR analyses were normalized to miR-16. Western blots were performed with antibodies as listed (Supplementary Table 1).

**Migration and invasion assays**. Transwell migration (8 μM inserts; BD-Biosciences) and invasion (BD-Biosciences; #354480) assays were performed for 6 h (human cells) and 16 h (mouse cells) using standard protocol[7]. Inserts were stained with crystal violet and migrated or invaded cells were analyzed and counted using Image-J software.

**3D culture assay**. Cells were seeded in a matrix comprising of matrigel (BD-Biosciences; #354230) or a mixture of matrigel and collagen (BD-Biosciences; #354236) and incubated for 6 days with regular replenishment with complete

media containing 2% matrigel until formation of spheroids[7]. Spheroids were imaged and analyzed using an inverted microscope.

**In vivo tumor and metastasis experiments.** All animal experiments were reviewed and approved by the Institutional Animal Care and Use Committee at The University of Texas M.D. Anderson Cancer Center. Cells were subcutaneously injected in the flanks of syngeneic 129/Sv mice of 8–10 weeks age and observed for tumor growth for a period of 4–8 weeks. Upon euthanasia, metastatic nodules on the surface of lung lobes were counted. Lung tissue was fixed in 10% Formalin and then processed for sectioning followed by H&E staining.

**In vivo cathepsin inhibitor treatment study.** All animal experiments were reviewed and approved by the Institutional Animal Care and Use Committee at The University of Texas M.D. Either vector GFP or *TMEM106B* inducible cells were subcutaneously injected in the flanks of syngeneic 129/Sv mice. Mice were fed with 625 mg/kg of doxycycline feed for a period of 4 weeks after 48 h of implantation of the cells. Both the vector control and *TMEM106B* mice cohorts were divided equally for treatment with either the cathepsin inhibitor E64D (Aloxistatin, from Selleckchem, S7393) or the vehicle as placebo controls. E64D was solubilized in complete DMSO at 60 mg/ml. For in vivo dosing, the solubilized E64D was suspended at a final concentration of 5% DMSO using a solution of 30% PEG-400 with 5% Tween-80 in water. Mice from respective cohorts were either treated with a daily dose of E64D at 25 mg/kg body weight, or similar volume of the vehicle with 5% DMSO (as placebo control), by oral gavage (P.O.) route. After 4 weeks of treatment mice were sacrificed and primary tumor volume and lung metastasis were scored. Primary tumors and internal organs were collected for RNA isolation and processed for biochemical assays to measure cathepsin activity.

**Lysotracker staining and microscopy.** Cells were grown on a glass bottom dishes and induced for expression of indicated genes. Induced cells were incubated with Lysotracker (100 nM) (Life technologies) for 1–2 n h in a tissue culture incubator at normal culture conditions. Post staining cells were washed with PBS (twice) and counter stained with Hoechst33342 for 15 min followed by washing in PBS and incubated in imaging medium (Life technologies), before imaging in real time by confocal microscopy. Similarly, stained cells were trypsinized and staining intensity was quantified by flow cytometry or on a fluorescent plate reader (Labmate).

**Magic Red Cathepsin activity detection assay.** Cells were grown on a glass bottom dishes and induced for expression of indicated genes. Induced cells were stained with Magic Red Cathepsin assay kit (Immuno Chemistry technologies; as per manufacturer's protocol) for 1–2 h in a tissue culture incubator at normal culture conditions. Poststaining cells were washed with PBS (twice) and counter stained with Hoechst33342 for 15 min followed by washing in PBS and incubated in imaging medium (Life technologies), before imaging in real time by confocal microscopy. Similarly, stained cells were trypsinized and staining intensity was quantified on a fluorescent plate reader (Labmate) using specified excitation/emission filter as mentioned in the kit.

**Cathepsin activity assays.** Cathepsin activity in conditioned media or conditioned PBS after lysosomal secretion was performed with specific cathepsin activity assay kits (Bio Vision, AbCam), as per manufacturer's protocol. Assay detection was performed with fluorescent plate reader (Labmate) at specified excitation/emission filter as mentioned in the kit.

**Ionomycin-mediated lysosomal exocytosis.** To assay for active cathepsins in the lysosomal compartment, lysosomal contents were secreted by inducing lysosomal exocytosis by treating a total of $5 \times 10^5$ cells with 10 mM Ionomycin when suspended in 1 ml PBS in presence of 2 mM calcium chloride ($Ca^{2+}$), which we referred as conditioned PBS. The conditioned PBS was assayed for activity of different cathepsins. To test whether active cathepsins released upon lysosomal exocytosis were essential to induce an invasive phenotype, Boyden chamber invasion assays were performed in presence or absence of protease inhibitors (Roche) or E64D (Selleckchem). Noninvasive 393P cells were suspended in conditioned PBS generated (as above) from $1 \times 10^6$ cells from different cells lines as indicated, in presence of 10 mM EGTA to chelate extra $Ca^{2+}$ in the buffer during the invasion assay. These assays were completed and invaded cells were quantified as described before[7].

**Immunofluorescence staining of *LAMP1*.** Cells grown on glass coverslips were first fixed using 4% paraformaldehyde for 10 min at 4 °C, followed by immune staining them for 1 h at 4 °C, using a PE conjugated *LAMP1* (BioLegend, PE anti-mouse CD107a (*LAMP1*)) primary antibody, without permeabilization, to specifically stain the *LAMP1* present on the cell surface. Coverslips were mounted in DAPI containing anti quench mounting solution and imaged by confocal microscopy. For quantification of staining, cells were trypsinized, fixed and stained as above and analyzed by flow cytometry.

**Cell fractionation.** Cell fractionations were performed using cell fractionation kit (Cell signaling Technologies) following manufacturer's protocol. Different fractions were estimated for protein content and western blot analysis was performed.

**LDH activity assay.** LDH activity assay was performed with Pierce LDH cytotoxicity assay kit (Thermofisher scientific) using manufacture's protocol. *TMEM106B* or GFP vector inducible cells were seeded in triplicates, in a 24 well plate at varying cell numbers and induced for 24 h. Conditioned media was used to test for cytotoxicity by measuring LDH activity.

**Data availability.** All relevant data, protocols and related information that are included in the article are available on demand from the authors.

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

## Acknowledgements

This work was supported by NCI K08 CA151661 (D.L.G.), an MD Anderson Cancer Center Physician Scientist Award (D.L.G.), a CDMRP Lung Cancer Research Program award W81XWH-12-10294 (D.L.G., K.L.S., and S.T.K.) and CPRIT grant RP120713. K.L.S. was supported by the NIH (U01CA168394). C.J.C. was supported by CPRIT grant RP120713 and NIH grant CA125123. D.L.G. is a Lee Clark Fellows of the University of Texas MD Anderson Cancer Center. We would like to thank members of the Gibbons lab and Scott Lab for assistance and critical reading of the manuscript.

## Author contributions

Study conceptualization, design and execution of project: S.T.K, K.L.S., and D.L.G. Data acquisition and statistical analysis: S.T.K., C.L.G., L.A.G., J.J.F., L.B.R., and C.J.C. Analysis, interpretation, and representation of data: S.T.K., C.J.C., K.L.S., and D.L.G. Manuscript writing, critical revision, and preparation of figures and tables: S.T.K., C.L.G., C.J.C., K.L.S., and D.L.G. Overall supervision and execution: S.T.K. and D.L.G.

## Additional information

**Competing interests:** The authors declare no competing interests.

