## [Peer Review File · Nature Communications]

Reviewers' comments:

Reviewer #1 (Remarks to the Author):

Remarks to the Author:

This manuscript claims that TMEM106B drives lung cancer metastasis. The findings are interesting and the studies are in general well performed. There are issues to address before this study reaches the rigor and significance suitable for publication in Nature Communications.

(1) The authors use a tumor xenograft flank model to search for mediators of lung cancer metastasis. This approach is hampered by in that the observations are not arising from a spontaneous lung cancer metastasis model. This needs to be experimentally addressed through additional autochthonous mouse models.

(2) How TMEM106B contributes to metastasis is not clear .

(3) The manuscript shows the use of KRAS mutant models but does not indicate whether the lysosomal signaling process identified is unique for KRAS mutant lung cancer or more general.

(4) The mechanism of TMEM106B and lysosomal-mediated metastatic potential is not elucidated. This is a major gap that limits the mechanistic depth the of the study.

Reviewer #2 (Remarks to the Author):

In this manuscript, the authors present TMEM106B as a novel candidate driving metastatic lung carcinoma. The authors show that TMEM106B overexpression upregulates the exocytosis of lysosomes. They find that this is caused by increased TFEB nuclear translocation. Importantly, it is shown that induction of lysosome exocytosis by TFEB activation promotes the release of cathepsins that favored cell migration and invasion. In addition, they showed that TMEM106B is highly expressed in tumors and it is a bad prognosis marker.

This is an interesting study and the effect of TMEM106B activation on lysosomal compartment and metastasis seems convincing. However, there are major concerns related with the role of TFEB and lysosomal exocytosis upon TMEM106B expression that should be addressed and take into account before the publication of this paper:

1) The novelty of this manuscript is basically the identification of TMEM106B as potential driver of metastatic lung cancer, and it is not clear at all whether this protein is the primary defect or a secondary effect of TFEB activation. Thus, TFEB is constitutively activated in some kidney cancers and in several pancreatic tumor cell lines (PMID:27668431; PMID:26168401). In addition, TFEB is able to induce lysosomal exocytosis (PMID:21889421). Thus, it would be necessary to determine what is first TFEB or TMEM106B? First, I would suggest testing whether TMEM106B is a transcriptional TFEB target. For example, by looking for E-box binding sites on TMEM promoter as well as to measure TMEM106B mRNA and protein levels upon TFEB overexpression. Moreover, since TFEB responds to a variety of lysosomal stressors (PMID:27252382) it is important to show whether TFEB is translocated by a specific mechanism of this observation is just a secondary effect of lysosomal stress due to TMEM106B overexpression. A simple experiment to investigate this important question could be the induction of TFEB nuclear translocation by different stimulus in normal cells or cells depleted of TMEM106B. In addition, the five candidate genes should be tested for their ability to induce TFEB nuclear translocation. This experiment is crucial to test whether the effect of TMEM106B on TFEB is selective.

2) Elevated secretion of lysosomal enzymes is indirect evidence of lysosomal exocytosis and could

also be alternatively explained. For example, the enhanced synthesis of LAMP (being a target gene for TFEB) might simply lead to an increase in the number of LAMP proteins that travel via the plasma membrane in route to lysosomes. Similarly, enhanced synthesis of lysosomal Cathepsins might, at the TGN, lead to a decreased sorting capacity into direct TGN-to-lysosome pathways, leading to direct secretion of proteins. Thus, it could be necessary to ask whether TMEM106B overexpression can induce conventional secretion TGN-to-PM. Moreover, the authors should provide direct and quantitative evidence for increased fusion of lysosomes with the plasma membrane. This could, for example, be achieved by the elevation of endogenous LAMP1 appearance in the PM by using immunofluorescence with antibodies against the luminal epitope of LAMP1, and by EM using the tannic acid method (see PMID:21889421).

In addition, some doubts about a direct role of TMEM106B inducing lysosomal exocytosis emerge from the experiments in figure 6. Thus, it is not clear whether TMEM106B overexpression by itself increase lysosomal exocytosis or its effects are more related with a very significant increase of lysosomal biogenesis and active cathepsins. The dramatic increase of lysosomal exocytosis upon ionomycin treatment compared with untreated TMEM expressing cells strongly suggest this alternative hypothesis.

3) The authors showed in figure 5A that TMEM106B overexpression dramatically increase RFP-LAMP1 staining on highly enlarged vesicles but also the staining reveal LAMP1 all around the cytoplasm and PM. Although authors reasoned that this result indicate an elevation of lysosomal exocytosis, to me it looks like a lysosomal swelling. Thus, I would suggest performing a similar experiment, but instead detecting endogenous LAMP1 upon the induction of TMEM106B (on inducible cell lines). Moreover, I would test LDH activity in the extracellular medium to discard TMEM106B-mediated toxicity.

4) TFEB induction promotes its own transcription (PMID:27252382). Thus, I would expect that similarly to the correlation with the 60% of clear genes (Figure 4E), TMEM106B expression in lung cancer samples should significantly correlates with an elevation of TFEB expression. Similarly, in figure 7, TFEB amplification should be queried to the TCGA data sets for lung adenocarcinoma. This is a very important question, since TFEB is upregulated in some tumors.

MINOR COMMENTS

- The ordering of the panels in Fig.S2 is confusing and does not coincide with the description of the results. Thus, FACs analysis of lysotracker red presented in S2B should be moved to the panel C together with result S2C6 of this figure

- In addition to TMEM106B-mediated induction of mCherry-TFEB in mouse cells, endogenous TFEB nuclear translocation should be tested in human H157 or HCC827 cell lines with inducible expression TMEM106B.

Response to reviewers' comments:

Reviewer #1 (Remarks to the Author):

This manuscript claims that TMEM106B drives lung cancer metastasis. The findings are interesting and the studies are in general well performed. There are issues to address before this study reaches the rigor and significance suitable for publication in Nature Communications.

We would like to thank the reviewer for their careful and critical analysis of the manuscript. To further solidify the data and hypothesis as proposed in the paper, we have performed the experiments suggested by the reviewer. We believe that the inclusion of these data has improved the manuscript.

(1) The authors use a tumor xenograft flank model to search for mediators of lung cancer metastasis. This approach is hampered by in that the observations are not arising from a spontaneous lung cancer metastasis model. This needs to be experimentally addressed through additional autochthonous mouse models.

We acknowledge the concerns raised by the reviewer and would like to clarify a few details regarding this point. The model used for the *in vivo* metastasis screen was a syngeneic mouse model. The putative candidate genes tested in the *in vivo* screen, which included TMEM106B, were selected based on their elevated expression in spontaneous metastases compared to primary lung tumors in the autochthonous $Kras^{LA1/+}/p53^{R172H\Delta G/+}$ mice and also in transcriptome comparisons of metastatic versus non-metastatic murine 344SQ and 393P syngeneic tumors, respectively, grown from cell lines isolated from spontaneous $Kras^{G12D}/p53^{R172H\Delta G}$ tumors. Therefore, the initial selection of TMEM106B as a candidate metastasis driver to test in the screen was based on its identification from a spontaneous lung cancer model.

Further, for the screen design we utilized a non-metastatic mouse lung cancer cell line (393P) established from a spontaneous $Kras^{LA1/+}/p53^{R172H\Delta G/+}$ tumor¹. Cell lines constitutively expressing each of the selected candidate ORFs were derived using the parental 393P. These cells expressing individual candidates were pooled and injected in the flanks of syngeneic host mice, allowing us to study their *in vivo* metastatic potential in a functional immune competent microenvironment, unlike that of the usual human/mouse xenograft flank model. The flank route of injection was consciously selected for several advantages, primarily including its conducive nature for high throughput *in vivo* screening with ease of monitoring the primary tumor growth and isolating functionally metastatic tumors from lung and other organs. Also, because of the intensive characterization of this syngeneic model in our lab and others over several years¹⁻³, the total time for the process of tumor formation and metastasis was well controlled and calibrated. We reasoned that this syngeneic flank model would provide a very low background and a high standard for any genes to produce *in vivo* metastasis to a secondary organ like lung. We subsequently validated the functional effects of TMEM106b in multiple different KP syngeneic models, with either ectopic expression or knockdown, as well as in multiple human NSCLC cell models, each of which yielded results consistent with the proposed mechanism.

Additionally, as the reviewer suggested, we have initiated experiments to confirm the role of TMEM106B as a driver of metastasis in an autochthonous lung model. We have delivered lentivirus expressing CRE and either GFP or TMEM106B (pHAGE-CRE-EF-TMEM106B / GFP) into the lungs of mice with conditional *K-ras*^{LSL-G12D/+} 4. These mice are being observed and imaged by CT over time for differences in tumor growth and incidence of metastasis. These autochthonous mouse experiments take upwards of 8 months for formation of visible tumors and metastasis and therefore the data is not included in the revision, as it is beyond the scope of our current study and well outside the time frame for the revisions requested by the journal. After completion of 8 months or signs of physical morbidity/sickness these mice will be sacrificed and examined pathologically for incidence of metastasis.

(2) How TMEM106B contributes to metastasis is not clear.

And

(4) The mechanism of TMEM106B and lysosomal-mediated metastatic potential is not elucidated. This is a major gap that limits the mechanistic depth the of the study.

We identified TMEM106B as a novel driver of lung cancer metastasis through an *in vivo* functional screen (co-submitted manuscript and Fig. 1-2 herein). We demonstrated that expression of TMEM106B results in production of enlarged lysosomes and these lysosomes are loaded with active cathepsins (Fig. 3-4). We further demonstrate that the enlarged and active lysosomes undergo exocytosis, in a calcium-dependent manner. Upon calcium influx, these lysosomes undergo lysosomal exocytosis, merge with the plasma membrane and release the lysosomal content of active cathepsins into the extracellular matrix (Fig. 5-6). We demonstrated that *in vitro* release of the lysosomal cathepsins is necessary and sufficient for TMEM106B-induced invasion.

Our results therefore outline the mechanism that elevated expression of TMEM106B in cancer cells enhances synthesis of enlarged lysosomes that are laden with bio-active hydrolases like cathepsins. These enlarged lysosomes undergo lysosomal exocytosis to release their active cathepsins into the extracellular matrix (ECM). Cathepsins and other lysosomal hydrolases are well documented in the literature to have potent functions in degradation of the ECM to create an invasive microenvironment, aiding in the dissemination and metastatic spread from the primary tumor.

To further address the concern raised by the reviewer and confirm whether elevated cathepsin synthesis and release was the major mechanism by which TMEM106B induces metastasis *in vivo*, we performed a therapeutic intervention study. Mouse cells with inducible expression of vector or TMEM106B were implanted into syngeneic mice. The mice were fed with doxycycline feed to induce expression of TMEM106B or GFP control. Half of the mice from either vector or TMEM106B-cell injected cohorts were randomized to treatment with a daily oral dose of 25 mg/kg body weight of the irreversible cysteine protease inhibitor E-64D (Aloxistatin) and the remaining half were dosed with the vehicle as control (Supp. Fig.S6A). After four weeks of treatment we observed that the primary tumors of the TMEM106B-cell injected mice had significantly more expression of TMEM106B (Fig. 6E and Supp. Fig. S6B). In the E64D-treated cohorts there was a complete suppression of the systemic cathepsins, as demonstrated by the

activity in the liver, and suppression of the Cathepsin B, K & H activity in the primary tumors (Fig. 6F-H). This demonstrated the effective systemic delivery of the drug, which would inhibit the activity of any systemically released cathepsins from the TMEM106B expressing primary tumors. Cathepsin-D is an aspartyl protease that was unaffected by E64D treatment, suggesting specificity of the inhibitor (Fig. 6I). Finally, we observed no appreciable difference in primary tumor sizes (Fig. 6J) between the treated and untreated mice, but the enhanced metastasis that was observed in the vehicle treated TMEM106B induced cohort was reversed to baseline in the E64D treated mice (Fig. 6K and Supp. Fig.S6C). These results demonstrate that TMEM106B can induce metastasis of lung cancer cells by the elevated synthesis and release of active lysosomal cathepsins, thereby creating an invasive microenvironment conducive for metastatic spread of cancer cells.

We have incorporated these new *in vivo* data into the revised version of the manuscript as indicated.

(3) The manuscript shows the use of KRAS mutant models but does not indicate whether the lysosomal signaling process identified is unique for KRAS mutant lung cancer or more general.

We identified that in NSCLC cells expression of TMEM106B enhances lysosomal synthesis and release of active cathepsins into the extracellular matrix, thus creating an invasive microenvironment that aids in the dissemination and metastatic spread of cancer cells. Since we identified TMEM106B in a Kras mutant mouse model we wanted to examine whether TMEM106B could produce a similar effect in wildtype Kras lung cancer cells. For this we made inducible TMEM106B expressing cell lines using lung adenocarcinoma cell lines with wildtype Kras gene. We used human HCC827 (WT Kras, mutant EGFR) and mouse Lewis lung carcinoma (LLC1-JSP) cell lines. We observed that in both HCC827 and LLC1-JSP cell lines induction of TMEM106B expression produced very similar cellular morphology and phenotype as seen in Kras mutant cells (Supp. Fig. S2-3 and S5A-B). Similarly, in analyses of TMEM106B copy number gain or amplification in human NSCLC clinical samples there was not a significant increase in mutant KRAS tumors (Supp. Fig. S5G). These results indicate that TMEM106B function as a driver of invasion and metastasis is independent of Kras mutational status.

We have incorporated the new data on these points into the revised manuscript in Supp. Fig. S2-3 and S5A-B.

Reviewer #2 (Remarks to the Author):

In this manuscript, the authors present TMEM106B as a novel candidate driving metastatic lung carcinoma. The authors show that TMEM106B overexpression upregulates the exocytosis of lysosomes. They find that this is caused by increased TFEB nuclear translocation. Importantly, it is shown that induction of lysosome exocytosis by TFEB activation promotes the release of cathepsins that favored cell migration and invasion. In addition, they showed that TMEM106B is highly expressed in tumors and it is a bad prognosis marker. This is an interesting study and the effect of TMEM106B activation on lysosomal compartment and metastasis seems convincing. However, there are major concerns related with the role of TFEB and lysosomal exocytosis upon TMEM106B expression that should be addressed and take into account before the publication of this paper:

We would like to sincerely thank the reviewer for the in-depth critical analysis of the manuscript and for suggestions to improve the data relating to the role of TFEB. We have performed all the experiments suggested by the reviewer, and hope that these results appropriately address their concerns.

1) The novelty of this manuscript is basically the identification of TMEM106B as potential driver of metastatic lung cancer, and it is not clear at all whether this protein is the primary defect or a secondary effect of TFEB activation. Thus, TFEB is constitutively activated in some kidney cancers and in several pancreatic tumor cell lines (PMID:27668431; PMID:26168401). In addition, TFEB is able to induce lysosomal exocytosis (PMID:21889421). Thus, it would be necessary to determine what is first TFEB or TMEM106B? First, I would suggest testing whether TMEM106B is a transcriptional TFEB target. For example, by looking for E-box binding sites on TMEM promoter as well as to measure TMEM106B mRNA and protein levels upon TFEB overexpression. Moreover, since TFEB responds to a variety of lysosomal stressors (PMID:27252382) it is important to show whether TFEB is translocated by a specific mechanism of this observation is just a secondary effect of lysosomal stress due to TMEM106B overexpression. A simple experiment to investigate this important question could be the induction of TFEB nuclear translocation by different stimulus in normal cells or cells depleted of TMEM106B. In addition, the five candidate genes should be tested for their ability to induce TFEB nuclear translocation. This experiment is crucial to test whether the effect of TMEM106B on TFEB is selective.

Our data indicates that TMEM106B expression could drive activation of TFEB-mediated transcription of TFEB target lysosomal genes (Fig. 4). To address the reviewers concern we first performed a motif search on the TMEM106B promoter (using JASPAR online tool: <http://jaspar.genereg.net/>), but were unable to identify any high scoring known TFEB binding motif on the TMEM106B promoter. This is consistent with published literature where TMEM106B was not identified as a TFEB transcriptional target (Sardiello, M et.al. Science, 2009⁵). Next, to confirm whether transient modulation of TFEB expression could affect TMEM106B levels, we either over expressed or knocked down the expression of TFEB and analyzed whether the expression changes affected TMEM106B expression. We observed that neither over expression nor knock down of TFEB had any significant effect on TMEM106B, either at the RNA (Supp. Fig. S4F) or protein (Supp. Fig. S4G) levels. These data and discussion are incorporated into the revised manuscript.

As suggested by the reviewer we performed experiments to test whether TMEM106B is critically required for TFEB activation via translocation to the nucleus. For this we stimulated nuclear translocation of TFEB in either control (scramble) cells or TMEM106B knockdown (Tmem106b-sh2) cells, by treating them with increasing concentrations of TORIN1 (which is a specific inhibitor of mTORC1 and previously shown to induce potent nuclear translocation of TFEB⁶⁻⁹). Upon TORIN1 stimulation we observed a clear increase in the nuclear abundance of TFEB in both the scramble control and the TMEM106B knockdown cells (Supp. Fig. S4D). These results indicate that TFEB nuclear translocation is not uniquely dependent on TMEM106B, although TMEM106B is one of several factors that can drive its translocation, and suggests that the TFEB activity may be due to increased lysosomal stress in the cancer cells.

Further when we performed cell fractionation assays with the 5 different cell lines (representing hits from the *in vivo* metastasis screen, as in Fig. 1A), we observed that only TMEM106B over expressing cells showed a considerable nuclear/cytoplasmic enrichment in comparison to the other hits and the mCherry control cells (Supp. Fig. S4C). These results indicate that TFEB nuclear translocation is specifically driven by elevated expression of TMEM106B and is not a common mechanism of the other *in vivo* metastasis drivers.

2) Elevated secretion of lysosomal enzymes is indirect evidence of lysosomal exocytosis and could also be alternatively explained. For example, the enhanced synthesis of LAMP (being a target gene for TFEB) might simply lead to an increase in the number of LAMP proteins that travel via the plasma membrane in route to lysosomes. Similarly, enhanced synthesis of lysosomal Cathepsins might, at the TGN, lead to a decreased sorting capacity into direct TGN-to-lysosome pathways, leading to direct secretion of proteins. Thus, it could be necessary to ask whether TMEM106B overexpression can induce conventional secretion TGN-to-PM.

We observed that with induction of TMEM106B there was a robust synthesis of enlarged lysosomal vesicles (Fig. 3A). Further when we stained the cells with LysoTracker we observed that these enlarged lysosomes showed strongly positive staining indicative of acidic lumens (Fig. 3A-C). These data indicate that the lysosomal lumens were filled with active acidic enzymes (lysosomal hydrolases). To confirm active lysosomal enzymes within the lysosomal vesicles which formed upon TMEM106B expression we stained the cells with Magic Red Cathepsin assays (Cathepsin B and K). These assays show positive staining only when the substrate is cleaved by the specific active Cathepsin. We clearly observed that all the enlarged lysosomal vesicles were strongly positive for the presence of active Cathepsin B and K (Fig. 3D-G and Supp. Fig. S3). This observation is best explained if the lysosomal enzymes like Cathepsin B&K are undergoing active and direct TGN to lysosome sorting and transport. These lysosomes undergo calcium-dependent exocytosis at the PM to release the cathepsins into the ECM (Fig. 5 and 6A-B). In addition to TGN-to-Lysosomal sorting there may be some additional cathepsins getting secreted directly through the TGN-to-PM route but we have not explored that possibility in this manuscript, as it will require a separate study of its own.

((2) Contd.) Moreover, the authors should provide direct and quantitative evidence for increased fusion of lysosomes with the plasma membrane. This could, for example, be achieved by the elevation of endogenous LAMP1 appearance in the PM by using immunofluorescence with antibodies against the luminal epitope of LAMP1, and by EM using the tannic acid method (see PMID:21889421).

To confirm that cells induced for TMEM106B expression undergo active lysosomal exocytosis the reviewer suggested we perform immune fluorescence surface staining for endogenous LAMP1 to detect its presence on the cell membrane. This was performed by first fixing the cells with paraformaldehyde, followed by immune staining them with a phycoerythrin (PE)-conjugated anti-LAMP1 primary antibody, without permeabilization, to specifically stain the LAMP1 present on the cell surface. We observed elevated surface staining of LAMP1 in TMEM106B expressing cells compared to control (Fig. 5C), which we also quantified by flow cytometric analysis of stained cells (Supp. Fig. 4H). These data demonstrate fusion of the lysosomal and cell membranes caused by increased lysosomal exocytosis. We appreciate this recommendation and have incorporated the new data into the revised manuscript.

(2) Contd.) In addition, some doubts about a direct role of TMEM106B inducing lysosomal exocytosis emerge from the experiments in figure 6. Thus, it is not clear whether TMEM106B overexpression by itself increase lysosomal exocytosis or its effects are more related with a very significant increase of lysosomal biogenesis and active cathepsins. The dramatic increase of lysosomal exocytosis upon ionomycin treatment compared with untreated TMEM expressing cells strongly suggest this alternative hypothesis.

We demonstrated in the conditioned media from cells which are induced for TMEM106B expression for 48-72 hours a significant increase in the presence of active cathepsins when compared to conditioned media from control cells. Also, the conditioned media from TMEM106B knockdown cells show a reduced abundance of active cathepsins compared to scramble control (Fig. 5D-G). These results indicate that TMEM106B could possibly drive lysosomal exocytosis. Additionally, lysosomes are calcium-dependent exocytic vesicles¹⁰. Hence, in the *in vivo* scenario, it is a definite possibility that in addition to a direct effect by TMEM106B, the enlarged, cathepsin-laden lysosomes synthesized upon elevated expression in TMEM106B, undergo exocytosis as a result of intracellular calcium fluxes that occur because of external (hypoxia) and internal stimuli (ER stress). Our results in Fig.6 demonstrate that upon acute induction of calcium flux, the active lysosomal cathepsin could be released in a few minutes and the released cathepsins are sufficient to drive cellular invasion through matrix. Since this assay was performed in a manner where the cells were suspended in the buffer for a matter of minutes, it is difficult to conclusively negate a role of TMEM106B in directly inducing lysosomal exocytosis. We have modified the Discussion to incorporate the possibility of both hypotheses to explain our results.

3) The authors showed in figure 5A that TMEM106B overexpression dramatically increase RFP-LAMP1 staining on highly enlarged vesicles but also the staining reveal LAMP1 all around the cytoplasm and PM. Although authors reasoned that this result indicate an elevation of lysosomal exocytosis, to me it looks like a lysosomal swelling. Thus, I would suggest performing a similar experiment, but instead detecting endogenous LAMP1 upon the induction of TMEM106B (on inducible cell lines). Moreover, I would test LDH activity in the extracellular medium to discard TMEM106B-mediated toxicity.

As suggested by the reviewer we performed surface LAMP1 staining by IF in the TMEM106B induced cells and observed a significant increase in surface staining by IF (Fig. 5C) which was quantified by flow cytometry (Supp. Fig. S4H). Using two different lung cancer cell line models (one mouse and one human), we also performed LDH activity assays with conditioned media from increasing number of cells with either induced expression of GFP or TMEM106B for 24 hours. We did not observe any difference in LDH activity between the cells, indicating a lack of toxicity due to TMEM106B expression (Supp. Fig. S4I). These data are incorporated in the revised manuscript.

4) TFEB induction promotes its own transcription (PMID:27252382). Thus, I would expect that similarly to the correlation with the 60% of clear genes (Figure 4E), TMEM106B expression in lung cancer samples should significantly correlates with an elevation of TFEB expression. Similarly, in figure 7, TFEB amplification should be queried to the TCGA data sets for lung adenocarcinoma. This is a very important question, since TFEB is upregulated in some tumors.

As suggested by the reviewer we first checked for differences in expression of TFEB upon TMEM106B expression in two different tumor cell models. We observed that there was no significant difference in TFEB levels in TMEM106B expressing cells when compared to controls (Supp. Fig. S5H). This was also consistent with our observation when we analyzed gene expression correlations from an extensive data set of more than 1000 human lung cancer samples. TMEM106B expression demonstrated no significant direct correlation with expression of TFEB (Fig. 4E). Further, as suggested by the reviewer, we queried the TCGA LUAD data set (TCGA-Provisional as queried using <http://www.cbioportal.org/>^{11,12}) to determine whether TFEB amplification could be associated as a prognostic indicator of disease outcome. We found the TFEB amplification was only present in 4% samples and did not correlate significantly with disease free survival (Supp. Fig. S5I). Our data suggests that in lung cancer cells, increased TMEM106B expression drives activation and nuclear translocation of TFEB, which results in upregulation of TFEB target lysosomal genes, without affecting TFEB transcription. These data are incorporated into the revised manuscript.

MINOR COMMENTS

- The ordering of the panels in Fig.S2 is confusing and does not coincide with the description of the results. Thus, FACs analysis of lysotracker red presented in S2B should be moved to the panel C together with result S2C6 of this figure.

We have made the suggested changes in the revised manuscript.

- In addition to TMEM106B-mediated induction of mCherry-TFEB in mouse cells, endogenous TFEB nuclear translocation should be tested in human H157 or HCC827 cell lines with inducible expression TMEM106B.

We performed the suggested experiment and observed a robust increase in the nuclear TFEB in human H157 cells induced for TMEM106B expression as compared to GFP induced control cells (Supp. Fig. S4A). These data have been incorporated into the revised manuscript.

References:

- 1 Gibbons, D. L. *et al.* Contextual extracellular cues promote tumor cell EMT and metastasis by regulating miR-200 family expression. *Genes & development* **23**, 2140-2151, doi:10.1101/gad.1820209 (2009).
- 2 Ahn, Y. H. *et al.* ZEB1 drives prometastatic actin cytoskeletal remodeling by downregulating miR-34a expression. *The Journal of clinical investigation* **122**, 3170-3183, doi:10.1172/JCI63608 (2012).
- 3 Yang, Y. *et al.* ZEB1 sensitizes lung adenocarcinoma to metastasis suppression by PI3K antagonism. *The Journal of clinical investigation* **124**, 2696-2708, doi:10.1172/JCI72171 (2014).
- 4 DuPage, M., Dooley, A. L. & Jacks, T. Conditional mouse lung cancer models using adenoviral or lentiviral delivery of Cre recombinase. *Nat Protoc* **4**, 1064-1072, doi:10.1038/nprot.2009.95 (2009).

- 5 Sardiello, M. *et al.* A gene network regulating lysosomal biogenesis and function. *Science* **325**, 473-477, doi:10.1126/science.1174447 (2009).
- 6 Martini-Stoica, H., Xu, Y., Ballabio, A. & Zheng, H. The Autophagy-Lysosomal Pathway in Neurodegeneration: A TFEB Perspective. *Trends Neurosci* **39**, 221-234, doi:10.1016/j.tins.2016.02.002 (2016).
- 7 Medina, D. L., Settembre, C. & Ballabio, A. Methods to Monitor and Manipulate TFEB Activity During Autophagy. *Methods Enzymol* **588**, 61-78, doi:10.1016/bs.mie.2016.10.008 (2017).
- 8 Napolitano, G. & Ballabio, A. TFEB at a glance. *J Cell Sci* **129**, 2475-2481, doi:10.1242/jcs.146365 (2016).
- 9 Settembre, C. & Ballabio, A. TFEB regulates autophagy: an integrated coordination of cellular degradation and recycling processes. *Autophagy* **7**, 1379-1381, doi:10.4161/auto.7.11.17166 (2011).
- 10 Rodriguez, A., Webster, P., Ortego, J. & Andrews, N. W. Lysosomes behave as Ca²⁺-regulated exocytic vesicles in fibroblasts and epithelial cells. *J Cell Biol* **137**, 93-104 (1997).
- 11 Gao, J. *et al.* Integrative analysis of complex cancer genomics and clinical profiles using the cBioPortal. *Sci Signal* **6**, pl1, doi:10.1126/scisignal.2004088 (2013).
- 12 Cerami, E. *et al.* The cBio cancer genomics portal: an open platform for exploring multidimensional cancer genomics data. *Cancer Discov* **2**, 401-404, doi:10.1158/2159-8290.CD-12-0095 (2012).

REVIEWERS' COMMENTS:

Reviewer #1 (Remarks to the Author):

The authors have addressed my questions and comments.

Reviewer #2 (Remarks to the Author):

Major concerns about the experimental approaches to study the role of TFEB and lysosomal exocytosis upon TMEM106B expression has been now addressed. Authors made a big effort to further investigated this process and now the results are convinced. Also minor points were addressed.

"REVIEWERS' COMMENTS:

Reviewer #1 (Remarks to the Author):

The authors have addressed my questions and comments.

Reviewer #2 (Remarks to the Author):

Major concerns about the experimental approaches to study the role of TFEB and lysosomal exocytosis upon TMEM 106B expression has been now addressed. Authors made a big effort to further investigated this process and now the results are convinced. Also minor points were addressed."

We are extremely thankful for the positive comments from the reviewers and are pleased to have satisfactorily addressed their questions and concerns in this revised version. We believe that incorporation of their suggestions has strengthened our manuscript. The data in the manuscript illustrates the mechanistic role of TMEM 106B as a driver of metastasis in lung adenocarcinoma and outlines how deregulation of lysosomal function and exocytosis facilitates invasion and metastasis. This work is based on extensive in vitro and in vivo studies with multiple preclinical models of lung adenocarcinoma, along with bioinformatics analysis of multiple large human datasets. As further demonstrated by the in vivo therapeutic treatment with aloxistatin, the data represented in this manuscript has the potential for development of clinical translation approaches to target the pro-metastatic effects of lysosomal dysregulation.